# Programming Every Example:
# Lifting Pre-training Data Quality Like Experts at Scale

**Fan Zhou** [* 1 2]  **Zengzhi Wang** [* 1 2]  **Qian Liu** [3]  **Junlong Li** [1]  **Pengfei Liu** [‡ 1 2 4]

## Abstract

Large language model pre-training has traditionally relied on human experts to craft heuristics for improving the corpora quality, resulting in numerous rules developed to date. However, these fixed rules lack the flexibility to address the unique characteristics of individual examples, yet crafting sample-wise rules is impractical for human experts. In this paper, we show that even small language models, with only 0.3B parameters, can exhibit substantial data refining capabilities. We propose Programming Every Example (PROX), a novel framework that treats data refinement as a *programming task*, and enables the model to refine corpora by generating and executing fine-grained operations, such as string normalization, for each individual example at scale. Experiments show that models trained on PROX-refined data consistently outperform other baselines across 10 benchmarks, demonstrating effectiveness across model sizes (up to 1.7B) and pre-training corpora (C4, RedPajama-V2, FineWeb, FineWeb-Edu, and DCLM). PROX also shows great potential in continual pre-training: on math domain, PROX boosts 7B models by up to 20% within 10B tokens—results typically achieved with much larger scale training (*e.g.*, 200B tokens). We believe PROX offers a way to curate high-quality pre-training data, and finally contributes to efficient LLM development.

## 1. Introduction

Large Language Models (LLMs) have made significant strides in capabilities (Meta, 2024; Achiam et al., 2023; Anthropic, 2024; Reid et al., 2024), excelling in tasks such as creative writing (Yuan et al., 2022), complex reasoning (Wei et al., 2022; Kojima et al., 2022), and agentic task planning and execution (Fan et al., 2022; Park et al., 2023). Behind these, massive, high-quality pre-training corpora form the backbone of these models, equipping them with the essential knowledge and reasoning abilities crucial for a wide range of downstream tasks (Penedo et al., 2024a).

The Internet offers vast amounts of data, but much of it is noisy and unrefined, requiring extensive cleaning and quality enhancement before being applied for pre-training. Previous works focus primarily on designing heuristic-based pipelines to lift data quality, such as document filtering (Rae et al., 2021; Penedo et al., 2024a; Soldaini et al., 2024) and perplexity-based scoring methods (Together, 2023), relying heavily on human expertise and manual adjustments (Zhang et al., 2024a). While widely adopted, these labor-intensive solutions are inherently limited by rule coverage and their inability to address every specific case. Recently, some efforts have explored leveraging LLMs for high-quality data acquisition. On the one hand, language models have been applied for data filtering or selection (Xie et al., 2023; Wettig et al., 2024; Yu et al., 2024; Dubey et al., 2024), but their role is largely limited to identifying low-quality documents without enabling fine-grained refinements (*e.g.*, string-level). On the other hand, LLMs are also being used to generate high-quality data directly, *i.e.*, data synthesis (Gunasekar et al., 2023; Li et al., 2023; Ben Allal et al., 2024). Unlike filtering, synthesis methods actively create or refine data to produce new documents, but they require substantial computational resources, limiting the methods' scalability. Despite the success, these methods can also inherit issues from LLMs like hallucination (Maini et al., 2024), and assessing their correctness and completeness in an interpretable manner remains a challenge (Liu et al., 2024a).

At this intersection of data processing efficiency and data quality improvement, we propose PROX, a model-based framework for pre-training-level data refinement. PROX focuses on refining corpora using smaller models at scale, offering a more efficient alternative. As shown in Figure 2, in practice, PROX first adapts small base language models (*e.g.*, < 1B) to data refining tasks through fine-tuning them on seed data. The refining models in PROX then determine

---

[*]Equal contribution  [1]Shanghai Jiao Tong University [2]Generative AI Research Lab (GAIR) [3]Sea AI Lab [4]Shanghai Artificial Intelligence Laboratory. Correspondence to: Pengfei Liu <pengfei@sjtu.edu.cn>.

*Proceedings of the 42nd International Conference on Machine Learning*, Vancouver, Canada. PMLR 267, 2025. Copyright 2025 by the author(s).

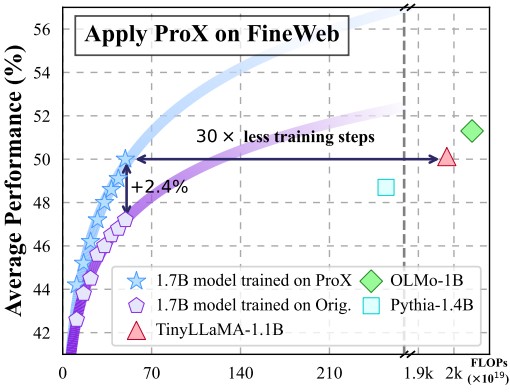 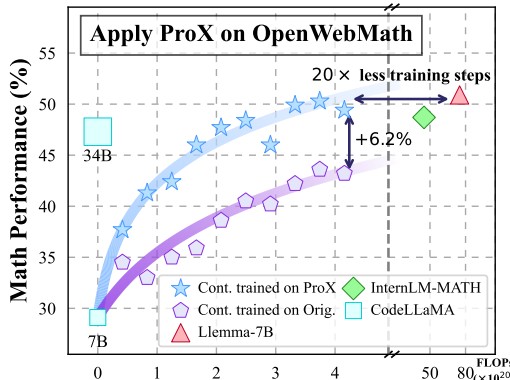

**Left**: Pre-training from scratch on general domain.    **Right**: Continual pre-training on math domain.

Figure 1: Training FLOPs vs. downstream performance. Although these corpora have been processed through expert-crafted rules, applying PROX still yields significant improvements over these baseline models trained with the original corpora. Moreover, models trained on PROX curated data achieve competitive performance with much fewer training FLOPs.

the appropriate operations for each document in the pre-training corpora via versatile programs, such as document filtering, string normalization and noisy line removal. The generated programs are then executed by a pre-defined executor, producing refined corpus ready for pre-training. In this way, PROX is empowered with language models to autonomously refine pre-training corpora, leveraging flexible function calls to enhance data quality.

Experimental results demonstrate that the proposed PROX framework consistently lifts data quality for **pre-training**. Specifically, PROX achieves an average improvement of $2.5\%$ over the original corpus on 10 downstream benchmarks and outperforms state-of-the-art data selection methods by over $1.0\%$ (§3.2). Furthermore, PROX demonstrates broad applicability across model sizes from 0.3B to 1.7B and achieves consistent performance gains across diverse pre-training corpora of varying quality, including RedPajama-V2 (Together, 2023), C4 (Raffel et al., 2020), FineWeb, FineWeb-Edu (Penedo et al., 2024a), and DCLM (Li et al., 2024) (§3.3). In domain-specific **continual pre-training**, as shown in Figure 1, PROX-refined OpenWebMath (Paster et al., 2024) boosts performance across 9 mathematical tasks (e.g., +20.3% on CODEL-LAMA-7B) while significantly enhancing efficiency, achieving similar downstream results with up to $20\times$ less computing (§3.4). Quantitative analysis suggests **scaling refinement FLOPs** achieves comparable performance with much lower training costs, offering an efficient path for LLM pre-training (§4).

## 2. Programming Every Example

### 2.1. Data Refinement Task Formulation

Given any document in the corpus $d \in \mathcal{D}$, such as an HTML extract or a textbook, we define data refinement as the pro-

cess of transforming $d$ into $\hat{d}$, where $\hat{d}$ exhibits higher quality. While it is challenging to formally define "higher quality" for pre-training data, we assume it can be described through qualitative improvements, such as the removal of advertisements, meaningless URL links, random code gibberish, and content lacking educational value, just as shown on the left side of Figure 2. Specifically, we formulate this refining process as the generation of a data processing program $\mathcal{Z}$, conditioned on $d$. The refined document $\hat{d}$ is then produced by executing program $\mathcal{Z}$ on the original document $d$. For instance, the "string normalization" can be a very fine-grained process transforming noisy strings into clean ones with executor $\mathcal{E}$ and program $\mathcal{Z}_{\text{normalize}}$:

$$\mathcal{E}(\mathcal{Z}_{\text{normalize}}, d) = (s'_i)_{i=1}^{|d|} \qquad (1)$$

where $s'_i = \text{normalize}(s_i)$ if $s_i$ needs normalization; otherwise, $s'_i = s_i$. Here, $d = (s_1, s_2, ..., s_{|d|})$ is the original document represented as a sequence of strings, and `normalize()` is our normalization function that maps certain strings to their normalized versions for simplicity. Moreover, document filtering is a special case of refining transformation, where executing $\mathcal{Z}_{\text{filter}}$ removes the entire document, i.e., $\mathcal{E}(\mathcal{Z}_{\text{filter}}, d) = \varnothing$. In this way, data quality improvements like cleaning or normalizing can be unified into standardized functions executed by $\mathcal{E}(\mathcal{Z}, d)$, where $\mathcal{Z}$ encodes function calls or heuristics for each specific task.

### 2.2. PROX Framework

**Overview** As shown in Figure 2, given any document $d$ as input, PROX utilizes the language model itself with parameter $\theta$ to generate the data refinement program $\mathcal{Z} = f_\theta(d)$. The snippet is executed within the executor $\mathcal{E}$, producing the refined document $\hat{d} = \mathcal{E}(f_\theta(d), d)$. We include two stages in the PROX framework, aiming to refine the data progressively, from rough to fine-grained. These two stages

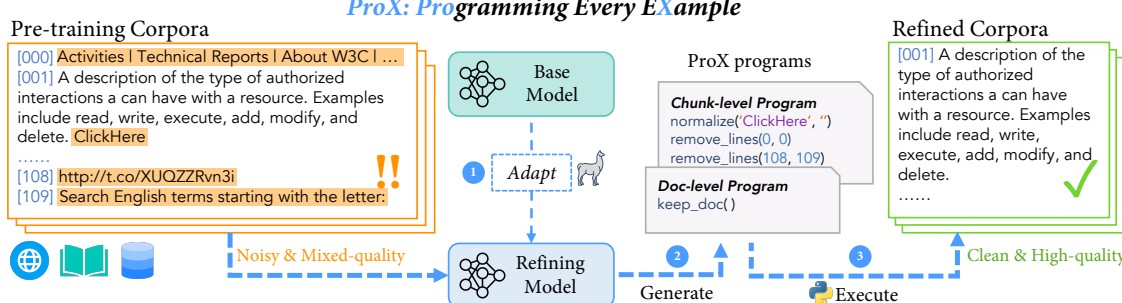

Figure 2: An overview of PROX framework: (1) adapt a base language model for data refinement; (2) generate elaborate programs for each document by PROX refining models, including document-level filtering and more fine-grained chunk-level refining; (3) execute the programs with the docs via a 🐍 Python executor, producing the refined high-quality corpora.

Table 1: PROX program design of document and chunk level refining stage. `doc` and `chunk` will also be sent into the corresponding functions as inputs for execution.

| Stage | Function Interface |
|---|---|
| Document Level | `drop_doc()` → <None> "*Delete the whole doc.*" |
| | `keep_doc()` → <str> "*Return the orignal doc.*" |
| Chunk Level | `remove_lines(line_start,line_end)` → <str> "*Delete noisy lines from chunk.*" ▷ `line_start`<int>, index of the first line to be removed ▷ `line_end`<int>, index of the last line to be removed |
| | `normalize(source_str, target_str)` → <str> "*Replace strings with normalized ones.*" ▷ `source_str`<str>, the noisy string pattern ▷ `target_str`<str>, the string for replacement |
| | `keep_chunk()` → <str> "*Return the orignal chunk.*" |

are referred to as *document-level programming* and *chunk-level programming*, as illustrated in Figure 2. In each stage, the PROX refining model will generate programs $\mathcal{Z}_{\text{doc}}$ and $\mathcal{Z}_{\text{chunk}}$ to refine corpora at varying levels of granularities.

**PROX Program Design**  Designing an effective program space is key to maximizing language model capabilities. For large-scale pre-training corpora, we prioritize: (1) lightweight models capable of recognizing specific patterns, and (2) simplicity and efficiency despite higher computational costs compared to heuristic-based pipelines. To balance functionality and resource constraints, we let models generate function calls without detailed implementations, ensuring effective document manipulation while maintaining coherence. We present the function definitions in Table 1, which also constitutes the program space of PROX.

The most fundamental operations we aim to perform on a document are deletion and replacement. In PROX, we incorporate these types of operations across different stages to refine the corpus at different granularities: (1) In the document-level programming stage, we define the functions `drop_doc()` to delete a document and `keep_doc()` to retain it. (2) At the chunk-level programming stage, we split lengthy documents into smaller chunks and apply fine-grained operations to them. These operations include delet-

ing specific lines with `remove_lines()` and replacing strings with `normalize()`, providing flexibility in modifying content rather than dropping the whole document. For high-quality chunks that require no modifications, we use the `keep_chunk()` function. As shown in Table 1, while the individual functions may seem straightforward, their design space is flexible and capable of expressing complex rules developed by human experts. We believe expert-crafted rules can be projected into the program space of PROX, demonstrating that our approach simplifies and enhances the rule creation process, offering more systematic and scalable refinement capabilities.

**PROX Execution**  During the execution stage, the executor $\mathcal{E}$ executes the generated program snippets $\mathcal{Z}$ to refine the document. PROX integrates Pythonic grammars, wrapping operations into functions with parameters, implemented in Python for later execution. For example, in Figure 2, the document contains noisy patterns such as navigation bars, meaningless HTML links, and page indexes. The refining model generates programs to remove these lines and patterns. At the document- and chunk-level programming stages, PROX uses two refining models to generate programs with various function calls (see Table 1). We believe this sequential approach ensures structured and effective refinement, first addressing larger document noise, and then focusing on finer-grained cleaning.

### 2.3. Model Adaptation for PROX

It is generally difficult for off-the-shelf models to directly generate perfect PROX programs. In fact, generating such custom API calls is relatively challenging even for the most powerful LLMs at the current stage (Zhuo et al., 2024). Thus, it is necessary to curate some seed data to adapt the model for these scenarios. Under such consideration, we employ advanced LLMs to annotate these operations via zero-shot and few-shot prompting, and then adapt our small models to these tasks by supervised fine-tuning (SFT). As presented in Figure 3, we first apply additive scale scoring

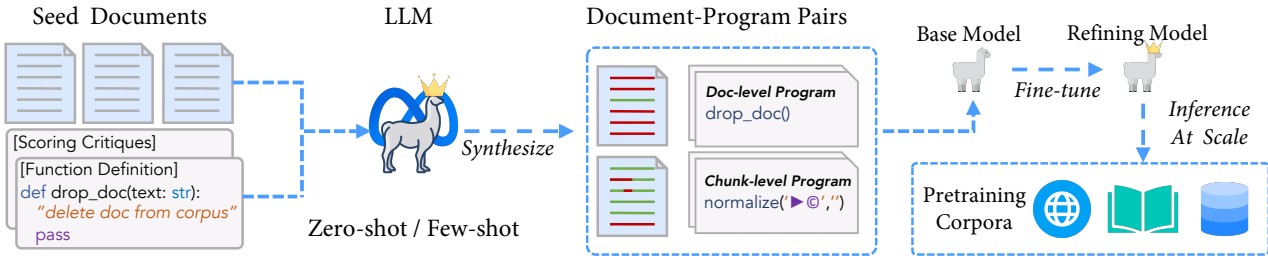

Figure 3: The illustration of the model adaptation in PROX. We employ powerful LLMs (LLAMA-3) to annotate random seed documents with valid programs and use *doc-program* pairs to fine-tune a small base language model, obtaining the refining model suitable for fine-grained data refining tasks.

prompts (Yuan et al., 2024; Penedo et al., 2024a), to split the corpus into kept and dropped documents, then use LLMs to annotate fine-grained programs for kept documents. Specifically, we leverage the LLAMA-3-70B model (Dubey et al., 2024) for seed data annotation, and the seed documents are randomly sampled from the original pre-training corpus. In PROX, this annotation is performed only once, and all models are adapted with the same curated data. To ensure the reliability of the collected data, we also conduct necessary checks for grammar correctness and control the removal ratio threshold. Detailed procedures for program synthesis and post-processing can be found in §A.1.

For simplicity and transparency, we use a small base language model (*e.g.*, 0.3B parameters) that we trained from scratch on approximately 26B tokens of original unrefined data, ensuring it is overtrained beyond the Chinchilla optimal points (Hoffmann et al., 2022). This model serves as the base model for ProX adaptation and also as the comparison baseline in subsequent experiments. The adapted models' performance will then be evaluated using the F1 score on the held-out validation dataset, both of which were derived from the seed data we collected earlier. We select the highest-performing checkpoints and employ them to generate programs $\mathcal{Z}$, for each document or chunk. These programs together with the documents are then executed using the corresponding function implementation, resulting in the final processed corpus. Please refer to the appendix for more training details (§A.2), implementation for calculating the F1 score (§A.3), and large-scale inference (§A.4).

## 3. Experiments

In this section, we first describe our experimental setup (§3.1), then verify the effectiveness of each PROX refining stage and compare it with various data selection methods tailored for pre-training (§3.2). We then apply PROX to various model sizes and corpora to demonstrate its broad applicability (§3.3). Finally, we apply PROX to the mathematical domain, showing its superiority in domain-specific continual pre-training (§3.4).

### 3.1. Experiment Setup

**Pre-training Corpora**   For the general domain, we begin with RedPajama-V2 (Together, 2023), a preprocessed large-scale dataset of 30 trillion tokens from diverse Internet sources, ready for pre-training. We further apply PROX on the C4 corpus (Raffel et al., 2020) with 198 billion tokens and the recent high quality datasets including FineWeb (as well as FineWeb-Edu) (Penedo et al., 2024a) and DCLM (Li et al., 2024). For specific domain experiments, we use OpenWebMath (Paster et al., 2024), a math-focused dataset with 15 billion tokens. Table 8 (§B.2) reports full details of our training corpus.

**Base Models**   Our experiments are conducted on various sizes of language models. **(1)** To verify different stages' effectiveness of PROX, we employ a 0.7B sized model sharing LLAMA-2 architecture (Touvron et al., 2023b), denoted as TLM-S, used for both pre-training from scratch and refining. We also compare PROX with data selection methods using PYTHIA-410 M/1 B's architecture (Biderman et al., 2023), as those employed in MATES (Yu et al., 2024). **(2)** For further evaluation of PROX using different refining and base model sizes, we scale model sizes from 0.3B (0.5×smaller, denoted as TLM-XS) to 1.7B (2×larger, denoted as TLM-M). **(3)** For domain-specific continual pre-training, we select TINYLLAMA-1.1B (Zhang et al., 2024b), LLAMA-2 (Touvron et al., 2023b), CODELLAMA (Rozière et al., 2023) and MISTRAL-7B (Jiang et al., 2023) as representative base models for their adequate training and solid performance. Detailed specifications and training recipes are provided in §B.3, especially in Table 9 and Table 10.

**Baselines**   To ensure a fair comparison within the same experiment, we maintain consistent training hyperparameters across most of the baselines, differing only in data refining and selection pipelines. We compare PROX with various baseline methods, including: **(1)** heuristic filtering rules used to create Gopher (Rae et al., 2021), C4 (Raffel et al., 2020), and FineWeb (Penedo et al., 2024a)), **(2)** fasttext-based filtering, trained on our PROX's seed data, **(3)** existing data selection techniques, including DSIR (Xie

Table 2: Detailed performance on 10 downstream tasks. All models use the same TLM-S architecture and are trained on RedPajama-V2. The doc-level (PROX-D) and chunk-level (PROX-C) refining are done by fine-tuning the raw data pre-trained model as a refining model. **Bolded** entries represent the best results. **#Win** represents the number of tasks where the method achieved the best performance.

| Method | ARC-C | ARC-E | CSQA | HellaS | MMLU | OBQA | PIQA | SIQA | WinoG | SciQ | AVG | #Win |
|---|---|---|---|---|---|---|---|---|---|---|---|---|
| Raw | 26.1 | 44.3 | 29.7 | 39.1 | 27.3 | 29.2 | 66.9 | 39.0 | 52.0 | 67.4 | 42.1 | 0 / 10 |
| Applying Rule-based filtering on Raw Data: GO = Gopher rules, C4 = C4 rules, FW = FineWeb rules. | | | | | | | | | | | | |
| GO | 25.7 | 44.0 | 31.3 | 40.2 | 27.3 | 29.0 | 66.3 | 39.0 | 51.2 | 68.9 | 42.3 | 0 / 10 |
| C4 | 25.0 | 46.0 | 31.0 | 40.5 | 27.1 | 29.2 | **68.5** | **40.5** | 51.7 | 66.6 | 42.6 | 2 / 10 |
| Fw | 25.2 | 46.8 | **32.6** | 39.6 | 27.2 | 29.0 | 66.5 | 39.4 | **52.4** | 69.2 | 42.8 | 2 / 10 |
| GO+C4+Fw | 25.2 | 43.9 | 30.0 | 41.9 | 27.5 | 31.0 | 67.0 | 39.9 | 51.9 | 65.3 | 42.3 | 0 / 10 |
| FASTTEXT | **26.9** | 49.9 | 29.5 | 39.0 | 28.5 | **31.8** | 64.7 | 39.6 | 52.1 | 70.4 | 43.3 | 2 / 10 |
| Applying PROX (ours) on Raw Data: D = Doc-level Programming, C = Chunk-level Programming. | | | | | | | | | | | | |
| PROX-D | 26.6 | 49.7 | 30.1 | 40.5 | **29.4** | 30.4 | 66.3 | 39.0 | 51.2 | 71.6 | 43.5 | 1 / 10 |
| PROX-D+C | 26.4 | **51.9** | 30.9 | **42.4** | **29.4** | **31.6** | 67.9 | 40.0 | 52.2 | **73.5** | **44.6** | **3 / 10** |

et al., 2023), DsDm (Engstrom et al., 2024), MATES (Yu et al., 2024), QuRating (Wettig et al., 2024), and **(4)** LLM synthesis approaches (such as INSTLM (Cheng et al., 2024) and COSMO (Ben Allal et al., 2024)) or LLM Pruning (SHEAREDLLAMA (Xia et al., 2024)). For domain-specific continual pre-training, we also compare with strong open-sourced models such as LLEMMA (Azerbayev et al., 2024), and INTERNLM2-MATH (Ying et al., 2024). Please refer to §C for full baseline details.

**Evaluation Setup** We compare the trained models' performance over a vast of datasets for comprehensive evaluation: (1) For general pre-training, we evaluate performance across ten selected tasks using lighteval's implementation (Fourrier et al., 2023); we have also included LM-eval-harness (Biderman et al., 2024) for fair comparison with data selection methods. (2) For domain-specific continual pre-training evaluation, we integrate nine mathematical related tasks and report few-shot chain-of-thought (CoT) (Wei et al., 2022) performance. The selected evaluation benchmarks, number of evaluation examples, and full details can be found in §D.

## 3.2. Verifying PROX's effectiveness

**Verifying Effectiveness for Each PROX Operation** We first conduct a series of experiments to verify the effectiveness of each PROX operation. We begin by training TLM-S on the RedPajama-V2 raw data for approximately 26B tokens (or 12.5K steps) as the initial baseline. Following Wettig et al. (2024) and for convenience, we sequentially apply the document-level and chunk-level refining pipelines by fine-tuning the 0.7B model itself. We then perform large-scale program synthesis and execution using the refining models, resulting in $\mathcal{D}_{doc}$ and $\mathcal{D}_{doc+chunk}$. Such 2-stage synthesis requires approximately 192 A100 GPU hours for processing 60B tokens of data. The downstream performance is presented in Table 2, including base models trained on the data produced by PROX refinement methods

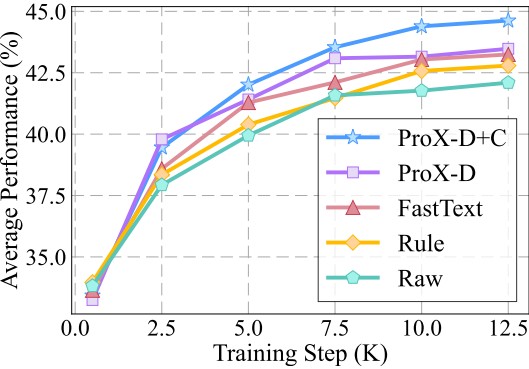

Figure 4: Downstream performance w.r.t. different training steps: first 0.5K, then evenly from 2.5K to 12.5K. Rule: the best performing FineWeb rule in Table 2.

and different rule-based baselines. Moreover, we visualize the dynamic benchmark performance in Figure 4, implying the consistent improvement of PROX over all baselines. See §E.1 for full detailed results of all intermediate checkpoints.

These results show that PROX is highly effective, outperforming the raw corpus with an average boost of 2.5%, including significant boosts such as 7.6% on ARC-E, and 3.3% on HellaSwag. Such improvements were achieved even on benchmarks that are typically prone to performance instability, such as SIQA, WinoGrande, and CSQA. By contrast, rule-based methods demonstrate relatively marginal overall improvement. For instance, Gopher rules achieve only a 0.2% boost, while C4 shows a modest 0.5% improvement. Furthermore, combining all three rules (as is done in constructing the official FineWeb corpus), does not lead to any larger enhancement in overall performance.

**Comparing with Data Selection Methods** Apart from comparing with heuristic methods, we also include existing representative model-based data selection methods tailored for pre-training corpora to verify PROX's effectiveness. In Table 3, we report both 0-shot and 2-shot performance un-

Table 3: Comparison with different data selection methods on 8 benchmarks using C4 corpus and PYTHIA architecture.

| Method | Total FLOPs (×1e19) | 0-shot | 2-shot | #Win |
|---|---|---|---|---|
| Model Architecture: PYTHIA-410M | | | | |
| Random | 6.4 | 42.7 | 43.8 | 0 / 8 |
| DSIR | 6.4 | 42.5 | 43.7 | 1 / 8 |
| DsDm | 10.7 | 43.4 | 44.1 | 0 / 8 |
| QuRating | 26.4 | 43.5 | 44.6 | 0 / 8 |
| MATES | 8.1 | 44.0 | 45.0 | 0 / 8 |
| PROX (ours) | 13.2 | **46.2** | **47.5** | **7 / 8** |
| Model Architecture: PYTHIA-1B | | | | |
| Random | 17.7 | 44.7 | 45.4 | 0 / 8 |
| MATES | 20.0 | 45.8 | 46.4 | 1 / 8 |
| PROX (ours) | 21.9 | **46.8** | **48.4** | **7 / 8** |

Table 4: Refining model's performance on valid set and token retention ratio of original corpus.

| Size | Doc-level | Chunk-level | Kept Ratio |
|---|---|---|---|
| XS (0.3 B) | 82.6 | 75.2 | 23.2% |
| S  (0.7 B) | 81.3 | 75.6 | 25.6% |
| M  (1.7 B) | 83.7 | 77.3 | 28.8% |

| Base Model Size | Raw | ProX-(xs) | ProX-(s) | ProX-(m) |
|---|---|---|---|---|
| xs | 39.6 | 42.3 | 41.9 | 41.9 |
| s | 42.5 | 43.9 | 44.6 | 43.5 |
| m | 43.4 | 46.0 | 46.2 | 45.7 |

Figure 5: PROX's effect over different model sizes.

der the same settings used in MATES (Yu et al., 2024). While we merely apply document-level stage (*i.e.*, PROX-D) which is indeed similar to data selection methods, we can see that PROX outperforms the strongest data selection method MATES, by 2.2% and 2.5% in 0-shot and 2-shot average performance for 410M model, and by 1.0% and 2.0% for 1B model. Additionally, PROX achieves the best performance on 7 out of 8 benchmarks tested, demonstrating its superiority over existing data selection methods. Full evaluation results are provided in Table 13 (§E.2).

### 3.3. Applying PROX across model sizes and corpora

In this section, we demonstrate that PROX can effectively benefit models beyond scales and across different corpora, and greatly improves the training efficiency.

**PROX works well across different scales.** We train a series of models from 350M to 1.7B (*i.e.*, TLM-XS, TLM-S, and TLM-M) on the same 26B tokens used in §3.2, and then fine-tune these models on doc-level and chunk-level tasks, obtaining refining models with different sizes. We then apply these models in both doc-level and chunk-level refining stages and use the curated data for from-scratch pre-training. We report the adaptation performance on refining tasks of different refining model sizes in Table 4. According to the validation performance, adaptation works well across all model sizes, all achieving > 80% F1 on document-level refinement, and > 75% F1 on chunk-level refinement. We further train models of different sizes from scratch using data produced by refining models of varying sizes. In Figure 5, the heatmap indicates that all refining models of three sizes improve data quality over raw data with a consistent performance boost of **2**% over all base model sizes. While TLM-XS curated data shows slightly better downstream performance, it has a lower token-level retention ratio (**23.2**% vs. **28.8**%) compared to larger models as reflected in Table 4. This implies that moderately larger models suggest a

favorable balance between data quality and quantity. These additional tokens likely provide more knowledge during pre-training without compromising downstream benchmark performance, showcasing an effective trade-off between data refinement and information preservation.

**PROX works well across pre-training corpora.** To assess the applicability of PROX across various pre-training corpora, we extend our experiments beyond RedPajama-V2 to C4 and the recently released top-quality corpus including FineWeb, FineWeb-Edu, and DCLM. We apply exactly the same PROX-xs refining models detailed in Table 4 to these corpora without constructing new seed data. We conducted larger-scale experiments by training 1.7B models from scratch for about 50B tokens, again achieving notable improvements: On ten downstream benchmarks, models trained on PROX's curated data showed improvements of +**2.0**% on RedPajama-V2, +**2.9**% on C4, +**2.4**% on FineWeb, +**0.9**% on FineWeb-Edu, and +**1.7**% on DCLM, as shown in Figure 6.

**ProX brings greater training efficiency.** To demonstrate the non-trivial nature of these results, we compared models trained on PROX curated data against various models trained by different approaches. These include models like TINYL-LAMA-1.1B (trained on 3T tokens, about **60**× of our training tokens and **40**× training FLOPs), SHEADLLAMA-1.3B (pruned from LLAMA-2-7B, with extra 50B tokens training), and models using LLM data synthesis, such as INSTRUC-TIONLM-1.3B and COSMO-1.8B. Our results, including TLM-M (PROX) and TLM-M (Raw), are presented alongside all these baselines in Figure 6. On FineWeb, which is recognized for its high-quality data, TLM-M using PROX-refined data performs comparably to pruned models like SHEADLLAMA-1.3B and TINYLLAMA-1.1B, despite their reliance on additional pruning techniques or much larger datasets. Moreover, using much less computing overhead for data refinement, our models even outperform models that

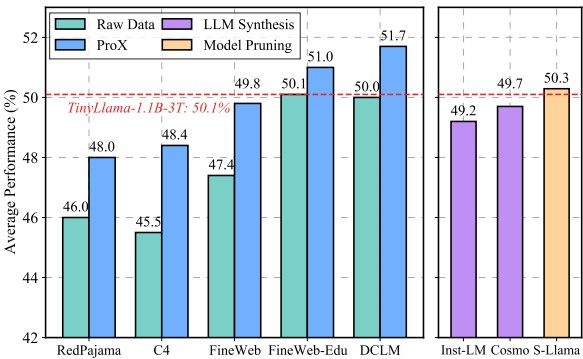

Figure 6: Performance of original data and PROX curated data trained models across different datasets using ≈ 50B tokens and comparison with existing models trained using different techniques like LLM data synthesis and direct model pruning. Inst-LM: INSTRUCTIONLM-1.3B; Cosmo: COSMO-1.8B; S-Llama: SHEAREDLLAMA-1.3B.

Table 5: Math domain continual pre-training (CPT) results with few-shot CoT evaluation performance averaged over 9 mathematical reasoning benchmarks.

| Model | Size | CPT | Method | Uniq Toks | Train Toks | Math AVG. Perf. |
|---|---|---|---|---|---|---|
| LLEMMA | 7B | ✓ | - | 55B | 200B | 50.9 (+21.8) |
| INTERNLM2 | 7B | ✗ | - | - | - | 36.1 |
|  | 7B | ✓ | - | 31B | 125B | 48.7 (+12.6) |
| TINYLLAMA | 1.1B | ✗ | - | - | - | 14.7 |
|  | 1.1B | ✓ | - | 15B | 15B | 21.5 (+6.8) |
|  | 1.1B | ✓ | Rule | 6.5B | 15B | 18.4 (+3.7) |
|  | 1.1B | ✓ | PROX | 5B | 15B | **25.7 (+11.0)** |
| LLAMA-2 | 7B | ✗ | - | - | - | 31.5 |
|  | 7B | ✓ | - | 15B | 10B | 42.8 (+11.3) |
|  | 7B | ✓ | PROX | 5B | 10B | **46.1 (+14.6)** |
| CODELLAMA | 7B | ✗ | - | - | - | 29.1 |
|  | 7B | ✓ | - | 15B | 10B | 43.2 (+14.1) |
|  | 7B | ✓ | PROX | 5B | 10B | **49.4 (+20.3)** |
| MISTRAL | 7B | ✗ | - | - | - | 51.6 |
|  | 7B | ✓ | - | 15B | 10B | 54.8 (+3.2) |
|  | 7B | ✓ | PROX | 4.7B | 10B | **59.2 (+7.6)** |

rely heavily on data synthesis with LLMs, underscoring the PROX's efficiency. Notably, models like INSTRUCT-LM-1.3B, trained on 100 billion tokens from a fine-tuned MIS-TRAL-7B synthesizer, and COSMO-1.8B, trained on 180B tokens (including 25B tokens synthesized by MIXTRAL-8x7B), require significantly more compute than PROX. In fact, their computational cost of data synthesis has far surpassed the training overhead.

### 3.4. Applying PROX to Specific Domains

We also demonstrate the potential of PROX in the continual pre-training scenario, specifically, in the mathematical domain. We apply the very same pipeline as in general domains to the OpenWebMath corpus (Paster et al., 2024), aiming to further mine and refine the high-quality and clean data from the crawled vast web pages. We apply PROX-xs series for refining as described in § 3.3 and adapt them on math seed data for the document-level and chunk-level refining tasks. Finally, about 5.5B tokens remain after document-level refining, and about 4.7B after chunk-level refining. We present the final mathematical evaluation results of models trained on OpenWebMath in Table 5, with comprehensive evaluation results and full ablation studies presented in §E.4.

**PROX boosts continual pre-training efficiency vastly.** Without any domain-specific design, Table 5 shows that pre-training on OpenWebMath refined by PROX brings 11.0% average performance improvements for TINYLLAMA-1.1B, 14.6% for LLAMA-2, 20.3% for CODELLAMA, 7.6% for MISTRAL, which clearly exceeds the improvements of all baselines, including their counterparts pre-trained on the original corpus. Notably, applying rule-based filtering does not improve performance; instead, it causes a 3.1% degra-

dation compared to continual pre-training on the original corpus. This suggests that universal heuristics are ineffective across all domains, highlighting the need for automated pipelines like PROX. Moreover, compared with existing math continual pre-training models like LLEMMA and INTERNLM2-MATH typically requiring hundreds of billions of training tokens, PROX demonstrates remarkable efficiency gains. A more controlled comparison further highlights this: LLEMMA-7B, based on CODELLAMA-7B, was trained on 200B tokens; whereas PROX, also starting from CODELLAMA-7B, reaches similar performance (50.9% vs. 49.4%) with just 10B tokens of training, indicating a **20×** reduction in training computes.

## 4. Analysis

**Impact on the Original Data** We compare the document's token length distribution of the original corpus with that of the PROX-refined corpus in Figure 7. In the general domain, the refined data exhibits a noticeable shift in the average number of tokens per document. Before refinement, documents with fewer than 100 tokens make up a significant portion of the corpus; after applying the PROX, the majority of them contain more than 200 tokens, with an average number of tokens per document increasing from 1217 to over 2000. This shift, however, is not observed in OpenWebMath (Middle part in Figure 7). One possible reason for this outlier is that the OpenWebMath corpus is collected mostly from sources different from the general domain, *e.g.,* online forums like Stack Exchange, and academic publisher websites such as Arxiv. And noises of these sources can be quite different from general domains. Further analysis and case studies on these documents are provided in §F.1, §F.2, and §F.3.

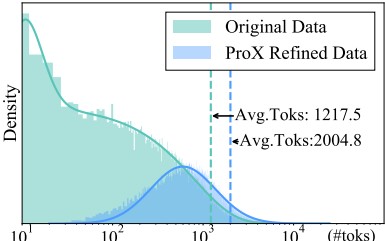 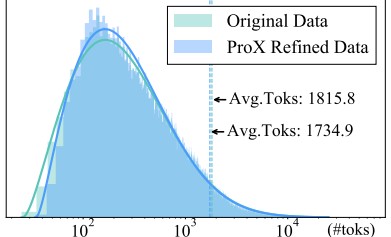 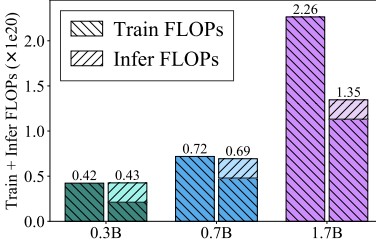

Figure 7: **Left** and **Middle**: Comparison of doc's token length distributions between original and PROX-refined data from RedPajama-V2 and OpenWebMath, respectively. **Right**: Total FLOPs comparison for achieving comparable downstream performance with/without PROX refining: 0.3B (Avg. Perf = 40.5), 0.7B (41.6), and 1.7B (42.9). [1]

Table 6: Analysis of programs generated by PROX: error program ratio and program complexity (function calls).

| Domain | Error Ratio | | Avg. Func Calls |
|---|---|---|---|
| | Doc-level | Chunk-level | Chunk-level |
| General | 0.04% | 0.36% | 3.7 |
| Math | 0.06% | 0.11% | 2.7 |

**PROX Generated Program Analysis**  PROX maintains extremely low failure rates ($< 0.4\%$) across both refining stages and domains while demonstrating computational efficiency, requiring only 3.7 and 2.7 average function calls for general and math domains respectively. This indicates PROX is well-suited for small models, achieving high reliability under resource constraints.

**Computing Overhead Analysis**  Although PROX demonstrates very promising results in downstream tasks, such large-scale inference will require an extra computing budget. Gladly, the relative computational cost for PROX will keep decreasing when developing larger models. In Figure 7, we calculate the FLOPs consumed by checkpoints with similar downstream performance, both with and without PROX. As model size increases, the proportion of inference FLOPs for applying PROX decreases. Surprisingly, for the largest 1.7B model, we achieve performance comparable to a model pre-trained on the original data, but with only $58\%$ of baseline's FLOPs. This demonstrates that refining methods like PROX not only enhance data quality but also provide efficiency for developing LLMs, reinforcing the value of allocating additional resources to refining pre-training data.

## 5. Related Works

**Pre-training Data Processing**  It has been a common practice to execute pre-processing before training due to the noisy nature of raw data from the Internet, which can hurt model performance (Touvron et al., 2023a; Together, 2023; Penedo et al., 2024a). The pipeline usually starts with document preparation, such as URL filtering and text extraction

(Wenzek et al., 2020; Smith et al., 2022). The remaining documents will then undergo several quality checks with heuristic rules like overall length, symbol-to-word ratio to determine whether they are kept, or aborted (Zhang et al., 2024a; Dou et al., 2024; Qiu et al., 2024). Finally, these documents are deduplicated with algorithms such as Min-Hash (Broder, 1997). In PROX, we use small language models for further data refining, outperforming heuristic rules with acceptable computational overhead.

**Data Selection Methods**  Data selection is more commonly applied in the later stages of large-scale data preprocessing. In supervised fine-tuning, it typically involves selecting a much smaller subset of samples while maintaining performance (Liu et al., 2024b). Recent efforts have extended these selection strategies to pre-training (Engstrom et al., 2024; Xie et al., 2023; Ankner et al., 2024; Sachdeva et al., 2024). Wettig et al. (2024) train a rater model to score documents on four quality criteria; MATES (Yu et al., 2024) apply LMs to estimate data influence and enables dynamic selection schema. Moreover, as mentioned in LLAMA-3 (Meta, 2024), LLAMA-2 (Touvron et al., 2023b) are used as text-quality classifiers that underpin LLAMA-3's training data. In PROX, we provide more fine-grained operations within documents for further improvements.

## 6. Conclusion

We introduced PROX, a framework that uses small language models to refine pre-training data through program generation and execution. Extensive experiments show that training on PROX curated data greatly improves performance on various downstream benchmarks, and holds effective across different model sizes and datasets. For the math domain, models trained on PROX curated data also yield significant improvements with $20\times$ less training tokens. Further analysis also shows applying PROX can achieve similar results with less compute for large-scale LLM pre-training. These results demonstrate PROX's potential to enhance data quality while reducing costs in training LLMs. We believe that PROX paves the way for developing more efficient LLMs, and scaling computing for data refinement may further accelerate progress in future exploration.

---

[1]The train FLOPs for the base model ($\approx 5.3 \times 10^{19}$) used to create the refining model are excluded. This is because any pre-trained LLM can theoretically serve as the base for refinement.

## Impact Statement

This paper presents work whose goal is to advance the field of Machine Learning. There are many potential societal consequences of our work, none which we feel must be specifically highlighted here.

## Acknowledgement

We extend our profound gratitude to Shanghai AI Lab and Sea AI Lab for generously providing valuable computational resources, which were instrumental in the realization of this project. Our sincere thanks also go to Mingxuan Wang and Jiaze Chen from ByteDance for their crucial support. We are deeply thankful to Ethan Chern from Shanghai Jiao Tong University and Yuqing Yang from University of Southern California for their early discussions and insightful contributions, and equally grateful to Zhoujun Cheng from UC San Diego, Yiheng Xu and Tianbao Xie from University of Hong Kong, and Terry Yue Zhuo from Monash University for their valuable feedback, to Guilherme Penedo and Loubna Ben Allal from Hugging Face for their guidance on hyperparameter tuning, to Zhibin Gou from Tsinghua University for providing advise on continual pre-training, to Lyumanshan Ye for helping with illustrations and color scheme design. Finally, special thanks go to Peiyuan Zhang from UC San Diego, representing the TinyLlama team, for providing a great open pre-training framework and supporting series of acceleration operators. These collective wisdom and unwavering support have been pivotal to our project. This project is supported by SJTU SEIEE - ByteDance Large Language Model Joint Laboratory, Shanghai Artificial Intelligence Laboratory.

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

# A. PROX Implementation Details

## A.1. Supervised Fine-tuning Data Collection

In this section, we elaborate on the detailed prompts used to generate the SFT data for model adaptation. In principle, We apply the same prompts for general domain corpora (including C4 (Raffel et al., 2020), RedPajama-V2 (Together, 2023), FineWeb (Penedo et al., 2024a)) and mathematical corpus (OpenWebMath (Paster et al., 2024)). All seed data is randomly sampled from the raw corpora.

**Document-level Programming**   We apply two zero-shot scoring prompts to evaluate and assign a combined score to each web document before synthesizing the (doc, program) pair. One of the prompts is the same as the one used in FineWeb-Edu, which is a prompt to let the model decide the educational score. Additionally in PROX, we add a new format scoring prompt, focusing on the format and structure of the document. Both prompts follow the additive style proposed by Yuan et al. (2024). Given these prompts, the language models generate short critiques and assign a score between 0 and 5.

In FineWeb-Edu, documents are retained only if the educational score (Edu Score) is greater than 2. However, this approach is too aggressive when attempting to preserve a larger portion of the tokens. For instance, FineWeb-Edu retains only 1.3 trillion tokens out of the original 15 trillion in the FineWeb corpus. To recall more documents, we relax the filtering criteria by incorporating the format score as follows:

$$\text{Filtering Criteria} = \begin{cases} \text{Edu Score} \geq 3, & \text{keep document;} \\ \text{Edu Score} = 2 \text{ and Format Score} \geq 4, & \text{keep document;} \\ \text{Edu Score} < 2, & \text{drop document.} \end{cases} \tag{2}$$

Finally, we use LLAMA-3-70B-INSTRUCT to annotate 51K data, splitting 5K for validation. [2]

The FineWeb-Edu prompt and our format scoring prompts are presented in Figure 8.

**Chunk-level Programming**   We apply chunk-level programming for more fine-grained operations. We find three very popular patterns that keep occurring in all corpus: (1) menu, navigation bars at the top of the document; (2) button, html elements, links; (3) footers.

In general, LLMs work well given within 5 few-shot examples. But to generate these program snippets more accurately, we apply few-shot prompting with LLAMA-3-70B-INSTRUCT for each type of noise. We merge these programs aiming to clean different types of noises, perform some grammar checking, and make them the final data for training and validation during the chunk-level refining stage. The annotated source comes from the same seed document used in the previous document filtering stage, accumulating to about 57K data, of which 5K is split as validation.

After the release of LLAMA-3.1-405B-INSTRUCT, We also try to use only one prompt aiming to remove all the noises. However, we find such practices lead to aggressive removal of the original document, often making the document less coherent. Finally, we decide to only keep the head part and tail part of the program generated by LLAMA-3.1-405B-INSTRUCT, which is previously mentioned in FinGPT (Luukkonen et al., 2023), and merge with the previous programs generated by LLAMA-3-70B-INSTRUCT.

The few-shot prompts used to generate program snippets are presented in Figure 9, Figure 10 and Figure 11.

**Comparison with FineWeb-Edu's Approach**   Compared with the recently released FineWeb-Edu, which also uses model-based scoring by applying a BERT model to evaluate documents, we find that our relaxed design retains more tokens without compromising overall data quality. Specifically, FineWeb-Edu retains about 1.3 trillion tokens out of a 15 trillion token corpus (less than 9%), while PROX curation typically keeps 23% to 28%, providing up to $\mathbf{3}\times$ more unique tokens for training.

---

[2]In the earlier stage of experiments, we found that a dataset of thousands of data points (i.e., 5K) is also sufficient to equip the model with the "programming" abilities. This generally holds true for both document-level and chunk-level programming tasks. Scaling the dataset size could enhance the model's robustness across various documents so we finally enlarge the pool to over 50K.

---

**Edu Scoring Prompts (Penedo et al., 2024a)**

Below is an extract from a web page. Evaluate whether the page has a high educational value and could be useful in an educational setting for teaching from primary school to grade school levels using the additive 5-point scoring system described below. Points are accumulated based on the satisfaction of each criterion:

- Add 1 point if the extract provides some basic information relevant to educational topics, even if it includes some irrelevant or non-academic content like advertisements and promotional material. - Add another point if the extract addresses certain elements pertinent to education but does not align closely with educational standards. It might mix educational content with non-educational material, offering a superficial overview of potentially useful topics, or presenting information in a disorganized manner and incoherent writing style. - Award a third point if the extract is appropriate for educational use and introduces key concepts relevant to school curricula. It is coherent though it may not be comprehensive or could include some extraneous information. It may resemble an introductory section of a textbook or a basic tutorial that is suitable for learning but has notable limitations like treating concepts that are too complex for grade school students.
- Grant a fourth point if the extract highly relevant and beneficial for educational purposes for a level not higher than grade school, exhibiting a clear and consistent writing style. It could be similar to a chapter from a textbook or a tutorial, offering substantial educational content, including exercises and solutions, with minimal irrelevant information, and the concepts aren't too advanced for grade school students. The content is coherent, focused, and valuable for structured learning.
- Bestow a fifth point if the extract is outstanding in its educational value, perfectly suited for teaching either at primary school or grade school. It follows detailed reasoning, the writing style is easy to follow and offers profound and thorough insights into the subject matter, devoid of any non-educational or complex content.
The extract:
¡EXAMPLE¿.
After examining the extract:
- Briefly justify your total score, up to 100 words.

- Conclude with the score using the format: "Educational score: ¡total points¿"

---

**Format Scoring Prompts**

Evaluate the provided web content extraction sample. Points are accumulated based on the satisfaction of each criterion:

0. Start with 0 points.
1. Add 1 point if the extract contains some readable content, even if it includes a significant amount of HTML tags, navigation elements, or other web page artifacts. The main content should be identifiable, albeit mixed with noise.
2. Add another point if the extract shows signs of basic cleaning. Most obvious HTML tags have been removed, though some may remain. The text structure begins to emerge, but non-content elements (e.g., footer links, button text) may still be present. The writing style may be disjointed due to remnants of page structure.
3. Award a third point if the extract is largely cleaned of HTML and most non-content elements. The main body of the content is intact and coherent. Some extraneous information (e.g., isolated URLs, timestamps, image alt text) may persist, but doesn't significantly impede readability. The extract resembles a rough draft of the original content.
4. Grant a fourth point if the extract is highly refined, with clear paragraph structure and formatting. Almost all HTML tags and non-content elements have been eliminated. Minimal noise remains. The content flows well and reads like a near-final draft, with consistent formatting and style.
5. Bestow a fifth point if the extraction is flawless. The content is entirely clean, preserving the original structure (paragraphs, headings, lists) without any HTML tags or web page elements. No extraneous information is present. The extract reads as if it were a professionally edited document, perfectly capturing the original content.
The extract:
¡EXAMPLE¿.
After examining the extract:
- Briefly justify your total score, up to 100 words.

- Conclude with the score using the format: "Extraction Quality Score: ¡total points¿"

---

Figure 8: Edu scoring prompts used in FineWeb (Penedo et al., 2024a) and newly proposed "format scoring" prompts for PROX.

Moreover, we conducted a preliminary study by training 0.7 billion parameter models on these data. We found that models trained on our curated data achieved similar downstream performance, as shown in Table 7. Therefore, we believe our current strategy is more suitable for large-scale pre-training, as it is capable of retaining more tokens while maintaining very high data quality.

Table 7: Comparing FineWeb-Edu with our strategy on TLM-S.

| Methods | Kept Ratio | ARC-C | ARC-E | CSQA | HellaSwag | MMLU | OBQA | PiQA | SIQA | WinoG | SciQ | AVG | #Win |
|---|---|---|---|---|---|---|---|---|---|---|---|---|---|
| FineWeb-Edu | 8.6% | 30.3 | 58.7 | 29.0 | 42.0 | 30.4 | 31.8 | 67.7 | 38.1 | 50.4 | 73.3 | 45.2 | 5/10 |
| FineWeb-PROX | **28.0%** | 27.7 | 55.7 | 30.4 | 44.2 | 29.5 | 31.0 | 68.8 | 39.3 | 52.2 | 72.8 | 45.2 | 5/10 |

**Navigation Removal Prompts**

You're tasked with generating Python programs to clean web text strings by removing navigation bars. The web text will be presented with line numbers starting from `[000]`. Your task is to use the following pre-defined functions to clean the text:

```python
def untouch_doc():
    """leave the clean doc untouched, for tagging clean and high quality doc."""

def remove_lines(start: int, end: int):
    """remove noisy lines from `start` until `end`, including `end`."""
```

Your goal is to identify navigation bars or menu items at the beginning of the text and remove them using the `remove_lines()` function. If the text doesn't contain a navigation bar or menu items, use the `untouch_doc()` function to indicate that no cleaning is necessary. If the line contains other text other than navigation, also call `untouch_doc` to escape overkilling. Here are some examples to guide you:
Example 1:

```
[doc]
[000] Home | Products | About Us | Contact
[001] Welcome to our website
[002] Here's our main content...
[/doc]
Program:
```python
remove_lines(start=0, end=0)
```
```

Example 2:

```
[doc]
341 US 479 Hoffman v. United States
341 US 479 Hoffman v. United States 341 U.S. 479
95 L.Ed. 1118
HOFFMANv.UNITED STATES.
Mr. William A. Gray, Philadelphia, Pa., for petitioner.
Mr. John F. Davis, Washington, D.C., for respondent.
......
[/doc]
Program:
```python
untouch_doc()
```
```

Example 3:

```
[doc]
[000]Police Search Tunbridge Wells House Over Human Remains Tip Off
[001]Posted: 16/04/2012 10:44 Updated: 16/04/2012 10:44 reddit stumble
[002]Crime, Body Buried In House, Buried Body, Buried Remains, Tip-Off, Uk News, Uk Police,
[003]Detectives are searching the gardens of a house following information that human remains may be
buried there.
[/doc]
Program:
```python
untouch_doc()
```
```

Example 4:

```
[doc]
[000]Home > Bollywood News > Bollywood Stars clash on Indian TV Bollywood Stars clash on Indian TV
[001]By Lekha Madhavan09:47 pm Betting big on the festive season, general entertainment channels (GECs)
are launching celebrity-driven shows, but media buyers are concerned about the audience split that is set
to happen.
[002]The fourth season of Bigg Boss on Colors is almost certain to clash with the fourth season of Kaun
Banega Crorepati (KBC) on Sony Entertainment Television (SET) in the second week of October.
[003]Another big property, Master Chef, to be hosted by Akshay Kumar, on STAR Plus, is also expected to go
on air in October. However, the channel is yet to disclose the launch date.
[004]Big-budget shows like these are often loss-making propositions for channels, as the operating cost is
very high and advertisement revenues do not suffice to cover the cost.
[005]Source: IBNS
[/doc]
Program:
```python
untouch_doc()
```
```

For each given web text, analyze the content and determine if there's a navigation bar or menu items at the beginning. If present, use `remove_lines()` or `normalize()` to remove them. If not, use `untouch_doc()` to indicate that no cleaning is needed.
Example: ¡EXAMPLE¿.
After examining the web text: - Briefly describe if the web extract contains navigation bar at the begining (10 lines).
- You must not mistakenly decide that title of the page is navigation bar and remove it.
- When the whole line is navigation bar, call `remove_lines`; if the line contains other information, call `normalize` to remove part of it.

- Give your program using the same format: ```python[your code]```

Figure 9: Few-shot navigation bar removal prompts.

## URL Removal Prompts

You're tasked with generating Python programs to clean web text strings by removing http lines. The web text will be presented with line numbers starting from `[000]`. Your task is to use the following pre-defined functions to clean the text:

```python
def untouch_doc():
    """leave the clean doc untouched, for tagging clean and high quality doc."""

def remove_lines(start: int, end: int):
    """remove noisy lines from `start` until `end`, including `end`."""

def normalize(source_str: str, target_str: str=""):
    """turn noisy strings into normalized strings."""
```

Your goal is to identify http links from the text and remove them using the `remove_lines()` or `normalize()` function. If the text doesn't contain http lines, use the `untouch_doc()` function to indicate that no cleaning is necessary.
Here are some examples to guide you:
Example 1:

```
[doc]
[013] http://groups.google.com/group/toowoombalinuxLast
[014] Breaking News: Major Event Unfolds
[015] http://code.google.com/p/inxi/
[/doc]
Program:
```python
# the whole line-[013] is http, so remove the line-[013]
remove_lines(start=13, end=13)
# the whole line-[015] is http, so remove the line-[015]
remove_lines(start=15, end=15)
```
```

Example 2:

```
[doc]
[000] The Impact of Climate Change on Global Ecosystems
[001] By Dr. Jane Smith
[002] Climate change continues to be a pressing issue...
[/doc]
Program:
```python
untouch_doc()
```
```

Example 3:

```
[doc]
[021]Bow-wow
[022]http://groups.google.com/group/toowoombalinuxLast edited by Puppyt on Mon 06 Jun 2011, 00:23; edited
1 time in total
[023]I would like to see something like Jitsi
[024]http://www.jitsi.org/. Plus some others incorporated into a puppy distro.
[/doc]
Program:
```python
# the http link in line 22 and line 24 comes with other text, so use normalize to ONLY remove the link
without touching text.
normalize(source_str="http://groups.google.com/group/toowoombalinuxLast", target_str="")
normalize(source_str="http://www.jitsi.org/.", target_str="")
```
```

For each given web text, analyze the content and determine if there's a navigation bar or menu items at the beginning. If present, use `remove_lines()` or `normalize()` to remove them. If not, use `untouch_doc()` to indicate that no cleaning is needed.
Example: ¡EXAMPLE¿.
After examining the web text: - do not remove text together with http.
- Briefly describe if the web extract contains http links; and make sure remove them will not influence the main content.
- Program only contain sequences of function callings and comments, no other codes.
- note line number starts with 0. make accurate annotations about line number. put the exact int line number of the given line. do not add 1 or minus 1.
- Give your program using the same format: ```python[your code]```

Figure 10: Few-shot URL removal prompts.

---

**Footer Removal Prompts**

You're tasked with generating Python programs to clean web text strings by removing footer sections, references. The web text will be presented with line numbers starting from `[000]`. Your task is to use the following pre-defined functions to clean the text:

```python
def untouch_doc():
    """leave the clean doc untouched, for tagging clean and high quality doc."""

def remove_lines(start: int, end: int):
    """remove noisy lines from `start` until `end`, including `end`."""

def normalize(source_str: str, target_str: str=""):
    """turn noisy strings into normalized strings."""
```

Your goal is to identify footer sections from the text and remove them using the `remove_lines()` function. Footers and references typically appear at the end of the text and may contain information such as copyright notices, contact details, or navigation links. If the text doesn't contain a footer section or any references, use the `untouch_doc()` function to indicate that no cleaning is necessary.
Here are some examples to guide you:
Example 1:

```
[doc]
[013] In conclusion, the study demonstrates significant findings.
[014] © 2023 Research Institute. All rights reserved.
[015] Contact: info@research-institute.com
[016] Follow us on social media: @ResearchInst
[/doc]
Program:
```python
# Remove the footer section starting from line 14
remove_lines(start=14, end=16)
```
```

Example 2:

```
[doc]
[000] The Impact of Climate Change on Global Ecosystems
[001] By Dr. Jane Smith
[002] Climate change continues to be a pressing issue...
[003] Further research is needed to fully understand its implications.
[/doc]
Program:
```python
untouch_doc()
```
```

Example 3:

```
[doc]
[020] Thank you for reading our newsletter.
[021] Stay informed with our latest updates!
[022] ---
[023] Unsubscribe | Privacy Policy | Terms of Service
[024] NewsletterCo, 123 Main St, Anytown, USA
[/doc]
Program:
```python
# Remove the footer section starting from the divider
remove_lines(start=22, end=24)
```
```

For each given web text, analyze the content and determine if there is a footer section or reference. If present, use `remove_lines()` to remove it. If not, use `untouch_doc()` to indicate that no cleaning is needed.
Example: ¡EXAMPLE¿.
After examining the web text:
- Briefly describe if the web extract contains a footer section or references; ensure that removing it will not influence the main content. If not, simply call `untouch_doc`.
- The program should only contain sequences of function calls and comments, no other code.
- Note that line numbers start with 0. Make accurate annotations about line numbers. Put the exact int line number of the given line. Do not add 1 or subtract 1.

- Give your program using the same format: ```python[your code]```

Figure 11: Few-shot footer removal prompts.

### A.2. Supervised Fine-tuning Details

**Training Parameters**   We use llama-factory (Zheng et al., 2024) as our main code base for the Adaptation Stage. We apply full parameter supervised fine-tuning on our base models: we train on the whole seed dataset for 3 to 5 epochs, with batch size as 64, and cosine learning rate schedular (lr from 1e-5 → 1e-6). Also, we find that the base model converges quite fast on these tasks, thus we do not apply further tuning over hyper-parameters, and keep the same training configurations for all the adaptation tasks.

### A.3. Evaluation Metrics for PROX Refining Tasks

**Document-level Refining Task**   The document filtering task is indeed equal to a binary classification problem, where documents are classified as either to be kept (1) or dropped (0). We evaluate the performance using the F1 score, calculated as follows:

$$\text{F1} = 2 \cdot \frac{\text{Precision} \cdot \text{Recall}}{\text{Precision} + \text{Recall}} \tag{3}$$

where:

$$\text{Precision} = \frac{\text{TP}}{\text{TP} + \text{FP}}, \quad \text{Recall} = \frac{\text{TP}}{\text{TP} + \text{FN}} \tag{4}$$

The F1 score ranges from 0 to 1 and we assume a higher F1 score indicates better classification performance.

**Chunk-level Refining Task**   This task actually contains two parts: line removal and string normalization. However, we find it rather hard to evaluate the normalization task, so we use the line removal accuracy to reflect the refining performance. We propose a line-wise F1 score metric:

The F1 score is computed by comparing the predicted noisy lines with the labeled noisy lines. First, we extract the noisy line indexes from both the prediction and the label. Then, we calculate the overlap between these two sets. The true positives (TP) are the number of lines in this overlap. False positives (FP) are the predicted noisy lines that are not in the labeled set, and false negatives (FN) are the labeled noisy lines that are not in the predicted set. The calculation is actually simple:

$$\text{TP (True Positives)} \quad = \quad |\text{Predicted Noisy Lines} \cap \text{Actual Noisy Lines}| \tag{5}$$

$$\text{FP (False Positives)} \quad = \quad |\text{Predicted Noisy Lines} \setminus \text{Actual Noisy Lines}| \tag{6}$$

$$\text{FN (False Negatives)} \quad = \quad |\text{Actual Noisy Lines} \setminus \text{Predicted Noisy Lines}| \tag{7}$$

Then we use same calculation of F1 score mentioned before, i.e., $\text{F1} = \frac{2 \cdot \text{TP}}{2 \cdot \text{TP} + \text{FP} + \text{FN}}$.

### A.4. PROX Inference at scale

Thanks to the Datatrove project (Penedo et al., 2024b), we are able to efficiently split, and load the whole corpus to each worker (which normally equals the number of GPUs since small models do not require tensor parallelism). We use the vllm (Kwon et al., 2023) to perform large-scale inference.

For chunk-wise programming, we will split the original document into several chunks, controlling the tokens of each chunk less than the context window. In practice, we normally replace the token count process with a word count process to save time and control the window size as $1,500$. The general algorithm is implemented as below:

---
**Algorithm 1** Document Chunk Splitting Algorithm

---
**Require:** Document $D$, context window size $W$
**Ensure:** Set of chunks $C$
1: $C \leftarrow \emptyset, c \leftarrow \emptyset$
2: **for** each line $l$ in $D$ **do**
3:     **if** TokenCount$(c + l) \leq W$ **then**
4:         $c \leftarrow c + l$             ▷ Add line to current chunk
5:     **else**
6:         **if** $c \neq \emptyset$ **then**
7:             $C \leftarrow C \cup \{c\}$             ▷ Save current chunk
8:         **end if**
9:         **if** TokenCount$(l) \leq W$ **then**
10:         $c \leftarrow l$             ▷ Start new chunk
11:         **else**
12:             $C \leftarrow C \cup \{\text{FlagAsSkipped}(l)\}$        ▷ Flag long line
13:             $c \leftarrow \emptyset$
14:         **end if**
15:     **end if**
16: **end for**
17: **if** $c \neq \emptyset$ **then**
18:     $C \leftarrow C \cup \{c\}$             ▷ Add the final chunk
19: **end if**
20: **return** $C$

---

# B. Pre-training Details

## B.1. Training Infrastructure

**Code Base**   Thanks to LitGPT (AI, 2023), and TinyLlama (Zhang et al., 2024b), we are able to flexibly train all our base models. We inherit several fused kernels from the TinyLlaMA, which is installed from the FlashAttention (Dao, 2024) including fused rotary positional embedding (RoPE) (Su et al., 2024), layer normalization, and cross-entropy loss to help saving memory. We mainly apply FSDP strategy (Zhao et al., 2023) to enable training larger scale models on multiple nodes.

## B.2. Pre-training Corpora

 Due to computing constraints and for fair comparison purposes, we cannot exhaustively train over the whole corpora. Thus, we apply random sampling for some of the pre-training corpora and make them as our pre-training data pools.

- For RedPajama-V2, We randomly download 70 file shards, obtaining a total data pool consisting about 500B tokens, we evenly separate it into 8 dumps, with each containing about 62.5B tokens; due to computing constraints, we use only 1 dump for verifying effectiveness (Section 3.2) and use 2 dumps for scaling the training to 50B tokens (Section 3.3);

- For C4, we download the whole dataset, which contains about 198B tokens;

- For FineWeb, we download the official 350B sample; [3]

- For OpenWebMath, we download the whole dataset.

We report the corpora details applied in each experiment in Table 8.

Table 8: The detailed breakdown for pre-training corpora in all experiments.

| Section | Experiments | Source | Data Description | Corpora Size (B) | Train Tokens (B) | Epoch |
|---|---|---|---|---|---|---|
| Section 3.2 | Table 2, Figure 4 | RedPajama-V2 | raw data size
after rule-based filtering
after PROX-D
after PROX-D+C | 62.5
31.5
19.0
16.0 | 26.2 | 0.42
0.83
1.38
1.64 |
| Section 3.2 | Table 3 | C4 | random
after PROX-D
other baselines | -
41.5 (GPT-NeoX)
- | 26.2 | -
0.63
- |
| Section 3.3 | Figure 5 | RedPajama-V2 | raw data size
after PROX-D+C (using PROX-xs)
after PROX-D+C (using PROX-s)
after PROX-D+C (using PROX-m) | 62.5
14.5
16.0
18.0 | 26.2 | 0.42
1.80
1.64
1.46 |
| Section 3.3 | Figure 6 | C4 | raw data size
after PROX-D+C (using PROX-xs) | 198.0
44.5 | 52.4 | 0.53
1.18 |
| | | RedPajama-V2 | raw data size
after PROX-D+C (using PROX-xs) | 123.5
29 | | 0.42
1.81 |
| | | FineWeb | raw data size
after PROX-D+C (using PROX-xs) | 79.0
18.0 | | 0.66
2.91 |
| Section 3.4 | Table 5, 1.1B model | OpenWebMath | raw data size
after rule-based filtering | 15.0
6.5 | 15.7 | 1.05
2.40 |
| | | | after PROX-D
after PROX-D+C | 5.5
4.7 | | 2.85
3.49 |
| Section 3.4 | Table 5, 7B model | OpenWebMath | raw data size
after PROX-D
after PROX-D+C | 15.0
5.5
4.7 | 10.5 | 0.70
1.91
2.23 |

---

[3] https://huggingface.co/datasets/HuggingFaceFW/fineweb/tree/main/sample/350BT

## B.3. Model Configuration and Training Parameters

Table 9: The details of the pre-training experiments' model architecture.

| Model | Hidden Size | Intermediate Size | Context Len | Heads | Layers | Vocab Size | # Params (w/o embed) |
|---|---|---|---|---|---|---|---|
| | | | Training From Scratch | | | | |
| TLM-XS | 1,280 | 2,048 | 2,048 | 16 | 24 | 32,000 | 354,284,800 (313,324,800) |
| TLM-S | 1,536 | 4,864 | 2,048 | 24 | 24 | 32,000 | 758,982,144 (709,830,144) |
| TLM-M | 2,048 | 8,192 | 2,048 | 32 | 24 | 32,000 | 1,741,785,088 (1,676,249,088) |
| PYTHIA-410M | 1,024 | 4,096 | 1,024 | 16 | 24 | 50,304 | 405,334,016 (353,822,720) |
| PYTHIA-1B | 2,048 | 8,192 | 1,024 | 8 | 16 | 50,304 | 1,011,781,632 (908,759,040) |
| | | | Continual Pre-training | | | | |
| TINYLLAMA-1.1B | 2,048 | 5,632 | 2,048 | 32 | 22 | 32,000 | 1,100,048,384 (1,034,512,384) |
| LLAMA-2-7B | 4,096 | 11,008 | 4,096 | 32 | 32 | 32,000 | 6,738,415,616 (6,607,343,616) |
| CODELLAMA-7B | 4,096 | 11,008 | 4,096 | 32 | 32 | 32,016 | 6,738,546,688 (6,607,409,152) |
| MISTRAL-7B | 4,096 | 14,336 | 4,096 | 32/8 (GQA) | 32 | 32,000 | 7,241,732,096 (7,110,660,096) |

Table 10: Training hyper-parameters of all base models.

| Model | Context Length | Batch Size | Max Steps | Warmup Steps | Weight Decay | Optimizer | LR Scheular | LR |
|---|---|---|---|---|---|---|---|---|
| | | | Training from Scratch | | | | | |
| TLM-XS | 1,024 | 2,048 | 12,500 | 500 | 0.1 | AdamW | cosine | 5e-4 → 5e-5 |
| TLM-S | 1,024 | 2,048 | 12,500 | 500 | 0.1 | AdamW | cosine | 5e-4 → 5e-6 |
| TLM-M | 1,024 | 2,048 | 12,500/2,500 | 500 | 0.1 | AdamW | cosine | 3e-4 → 3e-5 |
| PYTHIA-410M | 512 | 1,024 | 50,200 | 2,000 | 0.1 | AdamW | WSD | 1e-3 → 6.25e-5 |
| PYTHIA-1B | 512 | 1,024 | 50,200 | 2,000 | 0.1 | AdamW | WSD | 1e-3 → 6.25e-5 |
| | | | Continual Pre-training | | | | | |
| TINYLLAMA-1.1B | 2,048 | 1,024 | 7,500 | 0 | 0.1 | AdamW | cosine | 8e-5 → 8e-6 |
| LLAMA-2-7B | 4096 | 256 | 15,000 (early stop at 10,000) | 0 | 0.1 | AdamW | cosine | 8e-5 → 8e-6 |
| CODELLAMA-7B | 4096 | 1024 | 3,750 (early stop at 2,500) | 0 | 0.1 | AdamW | cosine | 3e-4 → 3e-5 |
| MISTRAL-7B | 4,096 | 256 | 15,000 (early stop at 10,000) | 0 | 0.1 | AdamW | cosine | 2e-5 → 2e-6 |

**Base Model Selection**   Our pre-training experiments are conducted using various sizes of decoder-only language models.

1. To verify different stages' effectiveness of PROX, we employ a 750M sized model sharing LLAMA-2 architecture (Touvron et al., 2023b), denoted as TLM-S, used for both pre-training from scratch and refining. We also compare PROX with data selection methods using PYTHIA-410M/1B's architecture (Biderman et al., 2023), as those employed in MATES (Yu et al., 2024).

2. For further evaluation of PROX using different refining and base model sizes, we scale the model sizes from 350M (0.5× smaller, denoted as TLM-XS) and 1.7B (2× larger, denoted as TLM-M), all based on the LLAMA-2 architecture.

3. For domain-specific continual pre-training, we select TINYLLAMA-1.1B (Zhang et al., 2024b), LLAMA-2 (Touvron et al., 2023b), CODELLAMA (Rozière et al., 2023) and MISTRAL-7B (Jiang et al., 2023) as representative base models for their adequate training and solid performance.

**Model Architecture**   The models we used in general and continual pre-training are presented at Table 9 with detailed architecture configuration.

**Training Hyperparameter Choice**   We primarily use a cosine learning rate scheduler and follow established settings used in Zhang et al. (2024b) and Lin et al. (2024). The default configurations for each experiment can be found below and we elaborate on full details in Table 10.

1. For general pre-training experiments, we set the learning rate to 5e-4 for TLM-XS and TLM-S, 3e-4 for TLM-M; the maximum sequence lengths are uniformly set to 2048, and the global batch size is set to 2M tokens.

2. Additionally, we align all our hyper-parameters with those used in MATES (Yu et al., 2024) to facilitate a direct comparison with their existing data selection methods, as previously shown in Table 3. In this case, we switch to the warmup-stable-decay (WSD) learning rate scheduler (Hu et al., 2024), as implemented in MATES. For a fair comparison with baselines implemented in MATES, we apply the exact same WSD Schedular (Hu et al., 2024):

$$lr(t) = \begin{cases} \frac{t}{W} \cdot \eta, & \text{if } t < W \\ \eta, & \text{if } W \leq t < S \\ 0.5^{4 \cdot (t-S)/D} \cdot \eta, & \text{if } S \leq t < S + D \end{cases} \tag{8}$$

where $W$ equals to 2000, $S$ equals to 50000, $D$ equals to 200.

3. For continual pre-training experiments, we set different hyperparameters for different base models, as shown in Table 10. We apply an early-stop mechanism mentioned in INTERNLM2-MATH (Ying et al., 2024) for 7B model experiments. We mainly refer to these settings to the setup reported in Rho-1 (Lin et al., 2024) and LLEMMA (Azerbayev et al., 2024). We do not use warmup in continual pre-training experiments.

## C. PROX Baseline Selection

To ensure a fair comparison w.r.t. training cost, we keep most of the training hyperparameters, such as training steps and batch size, consistent across baselines, with only the data refining and selection pipelines differing. We compare PROX to a series of baselines:

1. In § 3.2, to verify PROX's effectiveness, we first compare with PROX with regular pre-training over the raw RedPajama-V2 data. We also introduce heuristic baselines used to curate the FineWeb corpora, which is the combination of three filtering strategies from C4 (Raffel et al., 2020), Gopher (Rae et al., 2021), and newly crafted rules (as FineWeb rules). We also reproduce the fasttext classifier filtering which is reported as the strongest baseline in DCLM (Li et al., 2024) on our own corpus. Apart from rule-based baselines, we also introduce existing data selection techniques proposed in previous works, including (1) importance resampling: DSIR (Xie et al., 2023); (2) model-based selection: DsDM (Engstrom et al., 2024), MATES (Yu et al., 2024), and QuRating (Wettig et al., 2024).

2. In § 3.3, to test PROX on different model sizes and training corpora, we scale the TLM-M's training tokens to 50B over RedPajama-V2, C4, and FineWeb. To show PROX efficiency, we then directly compare with models covering a variety of pre-training approaches including (1) large-scale pre-training: TINYLLAMA-1.1B (Zhang et al., 2024b) trained on 3T tokens; (2) model pruning from existing models: (SHEADLLAMA (Xia et al., 2024) pruned from LLAMA-2 and trained on extra 50B tokens); (3) LLM synthesis (INSTRUCTIONLM-1.3B (Cheng et al., 2024) trained on MISTRAL-7B generated data and COSMO-1.8B (Ben Allal et al., 2024) trained on MIXTRAL-8x7B generated data).

3. In § 3.4's specific domain continual pre-training, apart from standard continual pre-training on TINYLLAMA-1.1B, LLAMA-2-7B, CODELLAMA-7B, and MISTRAL-7B, we additionally introduce with well-known and strong baselines trained on public (or partially public) data, including RHO-1 (Lin et al., 2024), INTERNLM2-MATH (Ying et al., 2024), LLEMMA (Azerbayev et al., 2024), and an internal checkpoint reported in DEEPSEEK-MATH (Shao et al., 2024).

# D. Downstream Tasks Evaluation

## D.1. General Pre-training Evaluation

**Lighteval Configurations**   We mainly borrow the evaluation benchmarks from FineWeb's nine selected "early signal" tasks (Penedo et al., 2024a), and use the implementation of lighteval (Fourrier et al., 2023) to test all our base models. We also introduce SciQ (Welbl et al., 2017) which is widely used in previous works and proved a good testbed (Mehta et al., 2024; Wettig et al., 2024). By default, we report the normalized zero-shot accuracy. All nine benchmarks are listed as below:

- ARC (Clark et al., 2018): including ARC-Easy (**ARC-E)** and ARC-Challenge (**ARC-C)**

- CommonSense QA (Talmor et al., 2019) (**CSQA**)

- HellaSwag (Zellers et al., 2019)

- MMLU (Hendrycks et al., 2021)

- OpenBook QA (Mihaylov et al., 2018) (**OBQA**)

- PIQA (Bisk et al., 2020)

- SocialIQA (Sap et al., 2019) (**SIQA**)

- WinoGrande (Sakaguchi et al., 2021) (**WinoG**)

- SciQ (Welbl et al., 2017)

We use the same configuration used in FineWeb's, which randomly picks $1,000$ samples for each dataset (for MMLU, it selects $1,000$ samples for each of the 57 subsets), and reports the normalized accuracy. This average performance is calculated over the nine benchmarks, where ARC-C and ARC-E are considered as two separate benchmarks, and MMLU is treated as a single benchmark. This approach differs slightly from the aggregation score calculation in FineWeb, as we believe MMLU's performance is relatively unstable, and we aim to give equal weight to all benchmarks, preventing MMLU from becoming a dominant factor. For the original lighteval scores, please refer to the §E.1, where we include a dynamic result curve that clearly illustrates the fluctuations in each benchmark.

We choose to present zero-shot evaluation mainly following settings used in all FineWeb's ablation experiments (Penedo et al., 2024a). We find the FineWeb evaluation maintains a very stable performance curve when training tokens gradually accumulate. Also, it is very time-efficient for fast evaluation regarding our extensive pre-training experiments(20+ final runs, with hundreds of intermediate checkpoints). We also present few-shot evaluation results in Table 11. Also, we find that not all benchmarks show better performance given few-shot prompts. For example, we do not observe a very clear performance boost on HellaSwag, MMLU, PIQA, and WinoGrande. Similar observation can also be noticed in recent works (Mehta et al., 2024; Muennighoff et al., 2023), where 0-shot Hellaswag and 0-shot WinoGrande show very close performances with 5-shot ones.

Based on these findings and considerations, we present zero-shot evaluation results in Table 2, Figure 4 and use it as our default evaluation metrics.

**LM-Eval Harness Configurations**   We also include the lm-evel-harness (Biderman et al., 2024) for zero-shot and few-shot performance, for fair comparison with different data selection methods including DSIR (Xie et al., 2023), DsDm (Engstrom et al., 2024), Qurating (Wettig et al., 2024) MATES (Yu et al., 2024). Similar to lighteval configuration, we include:

- ARC: including ARC-E and ARC-C

- HellaSwag

- LogiQA (Liu et al., 2020)

- OpenBook QA (OBQA)

- PIQA

- WinoGrande (WinoG)

- SciQ

We exclude the BoolQ (Clark et al., 2019) tasks from MATES (Yu et al., 2024), leaving eight tasks in total. This decision was made because we observed that the BoolQ benchmark performance exhibited severe fluctuations and showed a notable declining trend in the early stages. Therefore, we decided to exclude it from our evaluation set. Such a similar trend is also observed earlier in the OpenELM work (Mehta et al., 2024). We report both zero-shot and two-shot performance. If the metrics include *normalized accuracy*, we use that measure; otherwise, we use *accuracy*.

### D.2. Continual Pre-training Evaluation

We evaluate all benchmarks implemented in the math-eval-harness repository,[4] including:

- Math (**MATH**) (Hendrycks et al., 2021)

- GSM8K (Cobbe et al., 2021)

- SVAMP (Patel et al., 2021)

- ASDiv (Miao et al., 2020)

- MAWPS (Koncel-Kedziorski et al., 2016)

- MathQA (**MQA**) (Amini et al., 2019)

- TableMWP (**TAB**) (Lu et al., 2023)

- SAT MATH (Azerbayev et al., 2024)

We use few-shot CoT prompting (Wei et al., 2022) when evaluating these tasks, and report the accuracy of each task.

## E. Full Evaluation Results

### E.1. Detailed Performance on 10 Benchmarks in Sec 3.2

We report full evaluation results of checkpoints saved at different training steps in Section 3.2. We present the results for 0.7B models trained on data curated by different methods in Table 12, including models trained on raw data, rule-based filtered data, fasttext-filtered data, and data curated by PROX.

---

[4] https://github.com/ZubinGou/math-evaluation-harness

Table 11: Few-shot performance on 10 selected tasks. All models use the same TLM-S architecture and are trained on RedPajama-V2. The doc-level (PROX-D) and chunk-level (PROX-C) refining are done by fine-tuning the raw data pre-trained model as a refining model same as Table 2.

| Method | ARC-C | ARC-E | CSQA | HellaS | MMLU | OBQA | PIQA | SIQA | WinoG | SciQ | AVG |
|---|---|---|---|---|---|---|---|---|---|---|---|
| Raw | 25.5 | 50.3 | 33.2 | 39.9 | 27.8 | 29.2 | 67.8 | 38.7 | 52.4 | 71.5 | 43.6 |
| Rule-based | 26.2 | 50.9 | 34.1 | 41.8 | 27.8 | 29.2 | 66.8 | 40.5 | 52 | 72.8 | 44.2 |
| PROX-D | 29.1 | 55.7 | 35.6 | 41.8 | 29.4 | 29.2 | 66.8 | 38.3 | 51.3 | 77 | 45.4 |
| PROX-D+C | 27.2 | 59.9 | 38.3 | 42.8 | 29.7 | 31.4 | 67.1 | 40.3 | 50.2 | 75.8 | 46.3 |

Table 12: Full evaluation results on TLM-S.

| Train Steps | ARC-C | ARC-E | CSQA | HellaSwag | MMLU | OBQA | PiQA | SIQA | WinoG | SciQ | AVG |
|---|---|---|---|---|---|---|---|---|---|---|---|
| | | | | | Raw Data | | | | | | |
| 2500 | 22.1 | 39.0 | 27.6 | 31.6 | 25.9 | 26.6 | 61.2 | 37.3 | 48.9 | 59.1 | 37.9 |
| 5000 | 24.4 | 41.2 | 28.8 | 34.8 | 26.7 | 27.0 | 64.9 | 39.3 | 50.4 | 61.9 | 39.9 |
| 7500 | 26.5 | 43.9 | 29.5 | 37.2 | 27.2 | 29.0 | 64.8 | 38.7 | 50.8 | 68.2 | 41.6 |
| 10000 | 25.8 | 43.5 | 29.1 | 38.8 | 27.4 | 29.8 | 66.9 | 39.0 | 51.2 | 66.2 | 41.8 |
| 12500 | 26.1 | 44.3 | 29.7 | 39.1 | 27.3 | 29.2 | 66.9 | 39.0 | 52.0 | 67.4 | 42.1 |
| | | | | | Gopher | | | | | | |
| 2500 | 22.3 | 39.4 | 26.6 | 31.3 | 25.6 | 27.0 | 61.1 | 38.9 | 51.3 | 58.6 | 38.2 |
| 5000 | 25.1 | 41.4 | 29.8 | 34.3 | 26.4 | 27.2 | 64.5 | 39.6 | 52.1 | 62.9 | 40.3 |
| 7500 | 26.5 | 43.0 | 30.5 | 38.5 | 27.2 | 28.8 | 65.7 | 38.2 | 53.7 | 66.4 | 41.8 |
| 10000 | 26.2 | 44.2 | 31.8 | 39.2 | 27.5 | 29.4 | 66.6 | 38.9 | 51.3 | 68.2 | 42.3 |
| 12500 | 25.7 | 44.0 | 31.3 | 40.2 | 27.3 | 29.0 | 66.3 | 39.0 | 51.2 | 68.9 | 42.3 |
| | | | | | C4 | | | | | | |
| 2500 | 22.6 | 40.6 | 28.8 | 31.3 | 26.2 | 27.4 | 61.7 | 39.3 | 51.2 | 57.1 | 38.6 |
| 5000 | 22.9 | 41.6 | 29.3 | 36.0 | 26.8 | 27.6 | 64.7 | 40.2 | 50.9 | 63.6 | 40.4 |
| 7500 | 24.2 | 44.2 | 29.5 | 39.2 | 27.2 | 28.4 | 66.2 | 40.9 | 51.6 | 63.8 | 41.5 |
| 10000 | 24.6 | 44.8 | 30.4 | 39.5 | 27.0 | 29.4 | 68.7 | 40.9 | 51.7 | 63.9 | 42.1 |
| 12500 | 25.0 | 46.0 | 31.0 | 40.5 | 27.1 | 29.2 | 68.5 | 40.5 | 51.7 | 66.6 | 42.6 |
| | | | | | FineWeb | | | | | | |
| 2500 | 23.2 | 39.4 | 27.2 | 31.8 | 25.6 | 26.2 | 62.6 | 39.0 | 51.4 | 57.1 | 38.3 |
| 5000 | 24.2 | 42.3 | 29.8 | 36.2 | 27.0 | 28.4 | 64.3 | 38.9 | 51.4 | 61.4 | 40.4 |
| 7500 | 24.4 | 44.1 | 30.4 | 37.8 | 27.2 | 28.2 | 66.1 | 39.5 | 50.8 | 66.2 | 41.5 |
| 10000 | 23.6 | 46.6 | 32.0 | 39.6 | 27.0 | 27.8 | 66.3 | 39.2 | 53.1 | 70.5 | 42.6 |
| 12500 | 25.2 | 46.8 | 32.6 | 39.6 | 27.2 | 29.0 | 66.5 | 39.4 | 52.4 | 69.2 | 42.8 |
| | | | | | Gopher + C4 + FineWeb | | | | | | |
| 2500 | 23.6 | 39.3 | 27.6 | 32.1 | 25.8 | 26.0 | 61.7 | 39.8 | 50.9 | 55.4 | 38.2 |
| 5000 | 23.9 | 40.9 | 29.0 | 36.2 | 26.9 | 26.8 | 65.3 | 39.3 | 52.7 | 62.4 | 40.3 |
| 7500 | 25.6 | 42.2 | 30.7 | 39.7 | 27.0 | 28.4 | 66.0 | 40.2 | 51.8 | 60.9 | 41.2 |
| 10000 | 25.8 | 43.3 | 30.8 | 41.4 | 27.5 | 29.8 | 66.9 | 39.5 | 51.8 | 63.1 | 42.0 |
| 12500 | 25.0 | 43.9 | 30.0 | 41.9 | 27.5 | 31.0 | 67.0 | 39.9 | 51.9 | 65.3 | 42.3 |
| | | | | | PROX-D | | | | | | |
| 2500 | 25.6 | 43.2 | 27.7 | 32.9 | 27.2 | 27.0 | 61.3 | 39.4 | 50.6 | 63.0 | 39.8 |
| 5000 | 25.4 | 46.2 | 28.4 | 35.7 | 28.1 | 28.8 | 64.7 | 39.3 | 53.3 | 64.2 | 41.4 |
| 7500 | 26.9 | 49.2 | 29.1 | 39.2 | 28.6 | 30.8 | 65.4 | 38.8 | 51.2 | 71.7 | 43.1 |
| 10000 | 26.7 | 48.2 | 30.5 | 39.9 | 28.6 | 28.6 | 66.2 | 39.7 | 51.9 | 71.2 | 43.2 |
| 12500 | 26.6 | 49.7 | 30.1 | 40.5 | 29.4 | 30.4 | 66.3 | 39.0 | 51.2 | 71.6 | 43.5 |
| | | | | | PROX-D+C | | | | | | |
| 2500 | 24.9 | 43.4 | 27.3 | 32.1 | 26.9 | 28.2 | 60.9 | 38.8 | 51.2 | 60.8 | 39.5 |
| 5000 | 24.9 | 49.6 | 28.8 | 36.8 | 27.9 | 30.6 | 64.7 | 38.8 | 51.1 | 66.9 | 42.0 |
| 7500 | 25.5 | 51.2 | 30.8 | 38.8 | 28.4 | 31.2 | 67.3 | 40.2 | 50.3 | 71.7 | 43.5 |
| 10000 | 26.2 | 51.7 | 30.8 | 39.9 | 29.0 | 32.6 | 68.6 | 39.7 | 51.7 | 73.7 | 44.4 |
| 12500 | 26.4 | 51.9 | 30.9 | 42.4 | 29.4 | 31.6 | 67.9 | 40.0 | 52.2 | 73.5 | 44.6 |

## E.2. Detailed Performance on 8 Benchmarks used in Data Selection Experiments

The full benchmark performance used in data-selection method comparison experiments is presented in Table 13.

Table 13: Detailed evaluation results for different data selection methods.

| Method | ARC-C | ARC-E | HellaSwag | LogiQA | OBQA | PIQA | WinoGrande | SciQ | AVG |
|--------|-------|-------|-----------|--------|------|------|------------|------|-----|
| PYTHIA-410M 0-shot | | | | | | | | | |
| Random | 25.6 | 40.2 | 39.7 | 24.7 | 29.4 | 67.1 | 50.6 | 64.1 | 42.7 |
| DSIR | 23.8 | 39.9 | 39.6 | 27.0 | 28.4 | 66.8 | 51.5 | 63.1 | 42.5 |
| DsDm | 24.7 | 41.7 | 40.3 | **27.5** | 29 | 68.1 | 50.1 | 65.4 | 43.4 |
| QuRating | 25.4 | 42.0 | 40.7 | 25.3 | 30.2 | 67.5 | 52.1 | 64.8 | 43.5 |
| MATES | 25.0 | 41.8 | 41.0 | 25.7 | 30.8 | **68.7** | 52.7 | 66.0 | 44.0 |
| PROX | **27.2** | **48.9** | **43.1** | 26.9 | **31.8** | 68.4 | **54.1** | **69.5** | **46.2** |
| PYTHIA-410M 2-shot | | | | | | | | | |
| Random | 25.3 | 42.6 | 39.9 | 24.1 | 28.6 | 66.9 | 52.2 | 70.6 | 43.8 |
| DSIR | 23.6 | 42.0 | 39.8 | **26.1** | 28.6 | 66.1 | 51.6 | 71.4 | 43.7 |
| DsDm | 23.6 | 44.2 | 40.1 | 23.5 | 29.2 | 66.5 | 51.5 | 74 | 44.1 |
| QuRating | 23.6 | 43.9 | 40.4 | **26.1** | 30.2 | 67.4 | 51.4 | 74.1 | 44.6 |
| MATES | 25.3 | 43.8 | 40.6 | 24.9 | 30.6 | 67.1 | 53.4 | 74.1 | 45.0 |
| PROX | **27.0** | **52.7** | **42.6** | 23.7 | **32.8** | 68.2 | 53.9 | **78.9** | **47.5** |
| PYTHIA-1B 0-shot | | | | | | | | | |
| Random | 25.6 | 43.7 | 43.8 | 27.5 | 31.8 | 68.9 | 50.7 | 65.8 | 44.7 |
| MATES | 25.9 | 44.9 | 45.3 | **28.7** | **32.2** | 69.5 | 52.4 | 67.3 | 45.8 |
| PROX | **26.2** | **49.1** | **46.6** | 24.8 | **32.2** | 70.3 | **54.2** | **70.9** | **46.8** |
| PYTHIA-1B 2-shot | | | | | | | | | |
| Random | 25.5 | 45.1 | 42.9 | 24.6 | 30.0 | 68.3 | 52.1 | 74.6 | 45.4 |
| MATES | 26.8 | 46.1 | 44.8 | 25.2 | 30.6 | 68.7 | 51.6 | 75.7 | 46.2 |
| PROX | **27.3** | **54.5** | **46.2** | **26.6** | **32.2** | **69.0** | 53.9 | **77.4** | **48.4** |

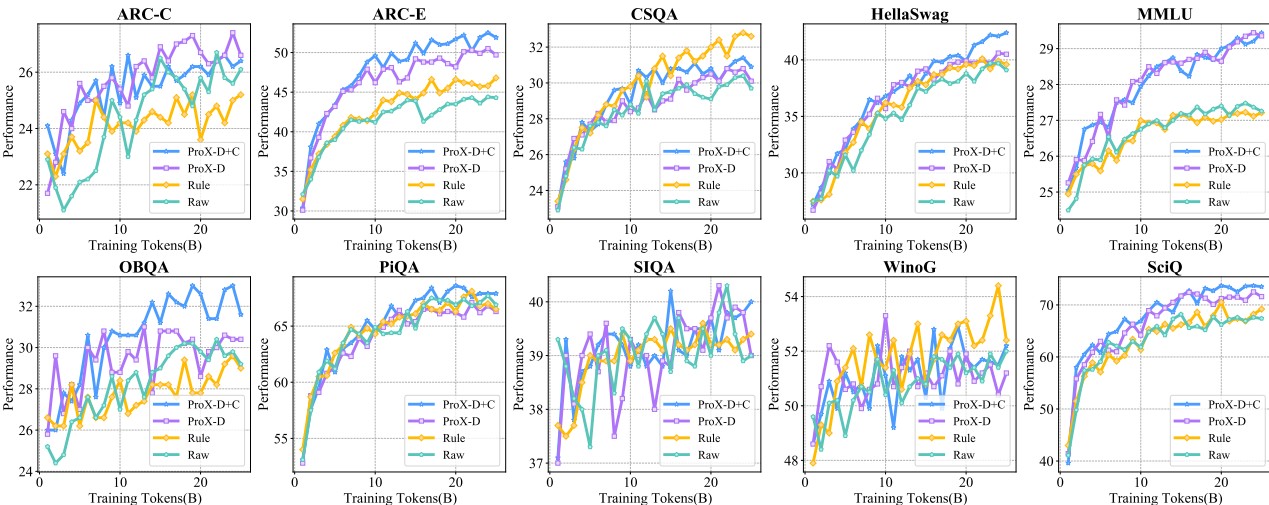

Figure 12: Visualization of dynamic performance on ten benchmarks. Rule: the best performing FineWeb rule in Table 2.

## E.3. Detailed Performance in Sec. 3.3

In § 3.3, we test PROX's effectiveness using different sizes of refining models, and also train a series of models by using these curated data. We report these detailed results in Table 14, Table 15 and Table 16.

Table 14: Full evaluation results of TLM-XS trained on different PROX model curated data.

| Train Steps | ARC-C | ARC-E | CSQA | HellaSwag | MMLU | OBQA | PiQA | SIQA | WinoG | SciQ | AVG |
|---|---|---|---|---|---|---|---|---|---|---|---|
| *TLM-XS trained on Raw data* | | | | | | | | | | | |
| 2500 | 22.5 | 38.5 | 27.0 | 29.1 | 25.8 | 25.0 | 60.2 | 38.8 | 50.4 | 58.6 | 37.6 |
| 5000 | 23.6 | 39.2 | 28.7 | 33.1 | 26.1 | 26.6 | 62.2 | 39.5 | 49.9 | 66.2 | 39.5 |
| 7500 | 23.8 | 42.7 | 28.0 | 33.4 | 26.0 | 26.2 | 64.0 | 39.3 | 51.5 | 67.0 | 40.2 |
| 10000 | 23.8 | 41.2 | 27.8 | 35.0 | 26.6 | 28.0 | 65.3 | 40.9 | 50.1 | 65.9 | 40.5 |
| 12500 | 22.6 | 41.9 | 29.7 | 32.8 | 26.2 | 26.4 | 62.2 | 39.3 | 51.3 | 63.3 | 39.6 |
| *TLM-XS trained on PROX-xs data* | | | | | | | | | | | |
| 2500 | 24.8 | 43.5 | 26.5 | 30.3 | 26.8 | 26.6 | 59.3 | 38.6 | 50.8 | 60.7 | 38.8 |
| 5000 | 23.7 | 44.3 | 28.1 | 33.8 | 27.3 | 28.8 | 61.3 | 38.9 | 50.9 | 70.2 | 40.7 |
| 7500 | 24.1 | 46.0 | 29.2 | 35.0 | 27.7 | 30.6 | 63.4 | 38.7 | 52.0 | 70.4 | 41.7 |
| 10000 | 25.3 | 46.1 | 28.3 | 35.7 | 28.1 | 29.2 | 64.4 | 38.5 | 51.2 | 70.6 | 41.7 |
| 12500 | 25.9 | 47.5 | 29.2 | 36.7 | 28.1 | 30.2 | 64.6 | 38.0 | 51.7 | 71.4 | 42.3 |
| *TLM-XS trained on PROX-s data* | | | | | | | | | | | |
| 2500 | 23.5 | 41.9 | 24.9 | 30.4 | 26.6 | 27.6 | 62.0 | 37.8 | 49.3 | 61.4 | 38.5 |
| 5000 | 24.7 | 44.5 | 27.0 | 33.8 | 27.5 | 28.0 | 62.4 | 38.0 | 50.6 | 67.0 | 40.3 |
| 7500 | 25.3 | 45.3 | 27.3 | 34.0 | 27.9 | 29.2 | 63.4 | 37.7 | 52.9 | 68.7 | 41.2 |
| 10000 | 25.6 | 45.7 | 27.6 | 35.6 | 28.6 | 30.2 | 63.6 | 37.4 | 52.0 | 71.1 | 41.7 |
| 12500 | 26.4 | 46.7 | 27.5 | 37.2 | 28.1 | 29.8 | 62.8 | 37.8 | 52.2 | 70.1 | 41.9 |
| *TLM-XS trained on PROX-m curated data* | | | | | | | | | | | |
| 2500 | 22.9 | 41.3 | 26.5 | 31.1 | 26.9 | 27.0 | 62.2 | 37.6 | 50.6 | 62.4 | 38.9 |
| 5000 | 25.8 | 44.0 | 27.3 | 34.0 | 27.1 | 29.6 | 63.1 | 38.5 | 51.8 | 64.9 | 40.6 |
| 7500 | 26.0 | 45.3 | 28.5 | 36.6 | 27.7 | 29.8 | 63.6 | 39.4 | 51.3 | 68.5 | 41.7 |
| 10000 | 26.0 | 46.6 | 28.8 | 37.3 | 27.6 | 30.6 | 63.3 | 38.7 | 51.6 | 70.3 | 42.1 |
| 12500 | 26.5 | 46.4 | 29.1 | 37.6 | 28.1 | 29.4 | 64.1 | 38.7 | 51.5 | 68.0 | 41.9 |

We also further scale PROX to other two pre-training corpora, C4 and FineWeb. We also scale our training to about 50B tokens, and directly compare with existing well-trained models developed by different research groups. We report our detailed results in Table 17, Table 18 and Table 19. We also present other models' results in Table 20.

Table 15: Full evaluation results of TLM-S trained on different PROX model curated data.

| Train Steps | ARC-C | ARC-E | CSQA | HellaSwag | MMLU | OBQA | PiQA | SIQA | WinoG | SciQ | AVG |
|---|---|---|---|---|---|---|---|---|---|---|---|
| TLM-S trained on Raw data | | | | | | | | | | | |
| 2500 | 22.1 | 39.0 | 27.6 | 31.6 | 25.9 | 26.6 | 61.2 | 37.3 | 48.9 | 59.1 | 37.9 |
| 5000 | 24.4 | 41.2 | 28.8 | 34.8 | 26.7 | 27.0 | 64.9 | 39.3 | 50.4 | 61.9 | 39.9 |
| 7500 | 26.5 | 43.9 | 29.5 | 37.2 | 27.2 | 29.0 | 64.8 | 38.7 | 50.8 | 68.2 | 41.6 |
| 10000 | 25.8 | 43.5 | 29.1 | 38.8 | 27.4 | 29.8 | 66.9 | 39.0 | 51.2 | 66.2 | 41.8 |
| 12500 | 26.1 | 44.3 | 29.7 | 39.1 | 27.3 | 29.2 | 66.9 | 39.0 | 52.0 | 67.4 | 42.1 |
| TLM-S trained on PROX-xs curated data | | | | | | | | | | | |
| 2500 | 23.8 | 44.1 | 26.5 | 33.5 | 26.9 | 29.4 | 60.7 | 38.9 | 50.6 | 62.1 | 39.6 |
| 5000 | 26.8 | 48.1 | 28.4 | 36.7 | 28.0 | 30.6 | 64.0 | 38.6 | 50.3 | 65.6 | 41.7 |
| 7500 | 26.9 | 49.0 | 30.6 | 39.5 | 28.2 | 29.6 | 65.3 | 39.6 | 52.2 | 69.6 | 43.0 |
| 10000 | 26.7 | 51.3 | 29.4 | 40.1 | 28.3 | 31.8 | 64.1 | 39.3 | 51.4 | 69.9 | 43.2 |
| 12500 | 26.8 | 52.1 | 30.2 | 41.8 | 28.5 | 31.6 | 65.5 | 39.5 | 51.9 | 70.8 | 43.9 |
| TLM-S trained on PROX-s curated data | | | | | | | | | | | |
| 2500 | 24.9 | 43.4 | 27.3 | 32.1 | 26.9 | 28.2 | 60.9 | 38.8 | 51.2 | 60.8 | 39.5 |
| 5000 | 24.9 | 49.6 | 28.8 | 36.8 | 27.9 | 30.6 | 64.7 | 38.8 | 51.1 | 66.9 | 42.0 |
| 7500 | 25.5 | 51.2 | 30.8 | 38.8 | 28.4 | 31.2 | 67.3 | 40.2 | 50.3 | 71.7 | 43.5 |
| 10000 | 26.2 | 51.7 | 30.8 | 39.9 | 29.0 | 32.6 | 68.6 | 39.7 | 51.7 | 73.7 | 44.4 |
| 12500 | 26.4 | 51.9 | 30.9 | 42.4 | 29.4 | 31.6 | 67.9 | 40.0 | 52.2 | 73.5 | 44.6 |
| TLM-S trained on PROX-m curated data | | | | | | | | | | | |
| 2500 | 25.3 | 45.3 | 27.5 | 32.2 | 26.7 | 27.0 | 62.4 | 38.7 | 50.6 | 60.8 | 39.6 |
| 5000 | 26.1 | 45.4 | 28.6 | 37.2 | 27.4 | 27.8 | 65.7 | 38.9 | 50.9 | 65.6 | 41.4 |
| 7500 | 27.1 | 47.5 | 30.6 | 41.0 | 28.6 | 29.2 | 66.8 | 39.3 | 51.1 | 69.9 | 43.1 |
| 10000 | 26.7 | 50.5 | 30.7 | 41.5 | 28.4 | 30.2 | 67.0 | 40.1 | 49.9 | 70.9 | 43.6 |
| 12500 | 27.4 | 50.7 | 30.6 | 42.0 | 28.8 | 30.2 | 67.4 | 39.4 | 48.8 | 70.1 | 43.5 |

Table 16: Full evaluation results of TLM-M trained on different PROX model curated data.

| Train Steps | ARC-C | ARC-E | CSQA | HellaSwag | MMLU | OBQA | PiQA | SIQA | WinoG | SciQ | AVG |
|---|---|---|---|---|---|---|---|---|---|---|---|
| | | | | TLM-S trained on Raw data | | | | | | | |
| 2500 | 23.5 | 41.5 | 27.5 | 32.9 | 26.4 | 25.2 | 62.1 | 39.4 | 51.5 | 65.1 | 39.5 |
| 5000 | 24.0 | 42.1 | 29.6 | 37.6 | 27.6 | 27.2 | 65.0 | 39.7 | 53.2 | 68.5 | 41.4 |
| 7500 | 24.3 | 44.9 | 28.9 | 39.3 | 27.8 | 27.6 | 66.4 | 40.4 | 51.3 | 69.2 | 42.0 |
| 10000 | 24.8 | 46.1 | 29.6 | 41.4 | 27.9 | 28.4 | 67.5 | 39.8 | 51.9 | 70.9 | 42.8 |
| 12500 | 26.3 | 46.8 | 29.0 | 43.2 | 28.3 | 27.8 | 68.2 | 40.5 | 50.7 | 72.5 | 43.3 |
| | | | | TLM-M trained on PROX-xs curated data | | | | | | | |
| 2500 | 24.9 | 49.6 | 26.5 | 34.0 | 27.3 | 30.4 | 61.8 | 37.9 | 51.3 | 65.1 | 40.9 |
| 5000 | 26.7 | 47.6 | 28.6 | 39.7 | 28.5 | 31.8 | 65.4 | 39.5 | 50.2 | 70.7 | 42.9 |
| 7500 | 27.5 | 52.1 | 30.4 | 41.8 | 29.6 | 31.8 | 67.6 | 39.6 | 51.7 | 75.2 | 44.7 |
| 10000 | 28.4 | 54.7 | 29.8 | 45.2 | 30.8 | 31.8 | 67.9 | 39.7 | 52.0 | 77.7 | 45.8 |
| 12500 | 28.8 | 54.2 | 29.7 | 46.5 | 30.9 | 31.8 | 68.2 | 39.9 | 51.3 | 78.3 | 46.0 |
| | | | | TLM-M trained on PROX-s curated data | | | | | | | |
| 2500 | 25.3 | 45.7 | 27.8 | 34.2 | 27.8 | 29.0 | 64.4 | 37.5 | 49.3 | 66.3 | 40.7 |
| 5000 | 26.1 | 49.0 | 28.8 | 40.2 | 29.2 | 30.8 | 65.6 | 39.0 | 50.5 | 71.2 | 43.0 |
| 7500 | 27.7 | 53.6 | 31.1 | 44.1 | 29.6 | 34.8 | 67.6 | 39.4 | 52.5 | 72.2 | 45.3 |
| 10000 | 27.2 | 54.0 | 31.5 | 45.1 | 30.3 | 33.8 | 67.7 | 39.7 | 52.9 | 74.2 | 45.6 |
| 12500 | 28.6 | 56.1 | 31.8 | 45.5 | 30.5 | 34.4 | 68.5 | 39.4 | 51.3 | 76.1 | 46.2 |
| | | | | TLM-M trained on PROX-m curated data | | | | | | | |
| 2500 | 24.7 | 44.1 | 25.9 | 34.8 | 27.4 | 27.8 | 62.9 | 38.9 | 49.2 | 67.0 | 40.3 |
| 5000 | 27.7 | 48.0 | 26.8 | 40.5 | 28.5 | 30.6 | 67.4 | 39.4 | 50.3 | 69.1 | 42.8 |
| 7500 | 26.7 | 51.9 | 26.7 | 42.9 | 29.3 | 31.4 | 69.1 | 40.3 | 50.4 | 73.3 | 44.2 |
| 10000 | 28.4 | 52.4 | 27.9 | 45.0 | 29.7 | 32.0 | 70.2 | 40.0 | 51.9 | 75.4 | 45.3 |
| 12500 | 28.3 | 53.7 | 28.4 | 45.9 | 30.1 | 33.8 | 70.6 | 41.1 | 52.3 | 72.5 | 45.7 |

Table 17: Full evaluation results on scaling pre-training to about 50B tokens on RedPajama-V2.

| Train Steps | ARC-C | ARC-E | CSQA | HellaSwag | MMLU | OBQA | PiQA | SIQA | WinoG | SciQ | AVG |
|---|---|---|---|---|---|---|---|---|---|---|---|
| TLM-M trained on RedPajama-V2 raw data. | | | | | | | | | | | |
| 2500 | 24.0 | 42.9 | 26.6 | 33.7 | 25.9 | 26.0 | 62.4 | 39.4 | 52.3 | 64.0 | 39.7 |
| 5000 | 24.3 | 45.9 | 26.4 | 37.4 | 27.0 | 27.6 | 64.1 | 39.7 | 49.5 | 66.2 | 40.8 |
| 7500 | 25.1 | 45.3 | 28.8 | 40.3 | 27.1 | 29.2 | 66.3 | 39.1 | 51.7 | 66.9 | 42.0 |
| 10000 | 25.8 | 49.3 | 31.5 | 42.5 | 28.0 | 28.8 | 66.7 | 39.6 | 51.5 | 74.0 | 43.8 |
| 12500 | 25.3 | 50.1 | 30.2 | 43.0 | 28.2 | 30.0 | 66.6 | 39.2 | 51.1 | 74.2 | 43.8 |
| 15000 | 26.2 | 50.3 | 31.2 | 44.3 | 28.8 | 28.4 | 68.2 | 39.8 | 51.7 | 76.2 | 44.5 |
| 17500 | 25.8 | 51.1 | 30.8 | 44.7 | 29.0 | 29.6 | 67.7 | 39.2 | 52.6 | 75.2 | 44.6 |
| 20000 | 26.7 | 52.5 | 31.7 | 47.2 | 28.6 | 30.4 | 69.0 | 39.6 | 53.0 | 78.2 | 45.7 |
| 22500 | 27.4 | 51.7 | 32.1 | 47.2 | 29.3 | 30.4 | 69.5 | 39.5 | 51.9 | 78.5 | 45.7 |
| 25000 | 26.9 | 51.4 | 32.4 | 47.3 | 29.3 | 32.2 | 69.7 | 39.6 | 52.1 | 79.1 | 46.0 |
| TLM-M trained on PROX refined RedPajama-V2 data. | | | | | | | | | | | |
| 2500 | 24.8 | 46.8 | 27.2 | 33.8 | 27.3 | 28.2 | 61.3 | 38.6 | 50.3 | 65.1 | 40.3 |
| 5000 | 26.9 | 49.3 | 28.5 | 40.1 | 28.0 | 30.6 | 66.2 | 39.7 | 50.2 | 70.1 | 43.0 |
| 7500 | 28.5 | 53.1 | 29.2 | 41.7 | 29.4 | 33.2 | 66.9 | 39.3 | 53.0 | 73.0 | 44.7 |
| 10000 | 28.2 | 53.5 | 30.1 | 43.6 | 29.8 | 31.6 | 68.4 | 39.6 | 52.0 | 75.3 | 45.2 |
| 12500 | 29.5 | 55.3 | 30.2 | 46.4 | 30.5 | 32.2 | 68.6 | 40.2 | 52.6 | 76.9 | 46.2 |
| 15000 | 30.0 | 57.1 | 30.2 | 47.6 | 30.9 | 33.0 | 69.5 | 39.8 | 52.2 | 77.8 | 46.8 |
| 17500 | 31.5 | 59.6 | 29.4 | 49.5 | 31.6 | 33.6 | 69.4 | 39.8 | 53.0 | 78.9 | 47.6 |
| 20000 | 31.2 | 61.2 | 29.4 | 50.4 | 31.4 | 35.2 | 70.6 | 40.1 | 53.7 | 79.6 | 48.3 |
| 22500 | 32.0 | 61.7 | 30.2 | 51.4 | 31.4 | 34.0 | 70.0 | 39.9 | 53.2 | 79.5 | 48.3 |
| 25000 | 31.1 | 60.7 | 29.8 | 51.0 | 31.7 | 33.2 | 70.9 | 39.2 | 53.3 | 79.1 | 48.0 |

Table 18: Full evaluation results on scaling pre-training to about 50B tokens on C4.

| Train Steps | ARC-C | ARC-E | CSQA | HellaSwag | MMLU | OBQA | PiQA | SIQA | WinoG | SciQ | AVG |
|---|---|---|---|---|---|---|---|---|---|---|---|
| TLM-M trained on C4 raw data. | | | | | | | | | | | |
| 2500 | 22.4 | 39.7 | 26.8 | 36.5 | 26.5 | 27.6 | 64.8 | 40.2 | 50.1 | 60.0 | 39.5 |
| 5000 | 23.9 | 42.9 | 27.5 | 42.3 | 27.1 | 29.6 | 68.2 | 39.6 | 50.3 | 66.6 | 41.8 |
| 7500 | 25.1 | 44.8 | 28.2 | 45.4 | 27.1 | 29.2 | 70.7 | 40.7 | 51.6 | 66.3 | 42.9 |
| 10000 | 25.5 | 46.0 | 32.3 | 48.2 | 27.9 | 31.6 | 71.1 | 39.7 | 52.3 | 67.6 | 44.2 |
| 12500 | 25.8 | 48.8 | 30.3 | 49.7 | 27.9 | 31.6 | 71.2 | 40.9 | 52.0 | 69.4 | 44.8 |
| 15000 | 26.9 | 48.0 | 28.2 | 50.5 | 28.5 | 31.4 | 71.9 | 41.1 | 51.4 | 69.7 | 44.8 |
| 17500 | 26.6 | 48.8 | 30.3 | 52.1 | 28.6 | 31.2 | 73.2 | 41.6 | 52.0 | 70.0 | 45.4 |
| 20000 | 26.3 | 50.1 | 29.7 | 52.5 | 28.5 | 32.6 | 72.3 | 41.7 | 52.3 | 71.0 | 45.7 |
| 22500 | 25.8 | 50.7 | 31.0 | 52.9 | 28.8 | 33.8 | 73.0 | 41.6 | 53.0 | 71.5 | 46.2 |
| 25000 | 25.3 | 48.8 | 30.1 | 52.4 | 28.8 | 32.2 | 72.0 | 40.6 | 53.6 | 71.7 | 45.5 |
| TLM-M trained on PROX refined C4 data. | | | | | | | | | | | |
| 2500 | 24.1 | 45.9 | 26.0 | 37.3 | 27.2 | 29.0 | 66.3 | 39.8 | 50.8 | 65.9 | 41.2 |
| 5000 | 27.3 | 50.0 | 26.6 | 42.4 | 28.6 | 33.8 | 68.1 | 40.5 | 53.0 | 71.9 | 44.2 |
| 7500 | 28.3 | 53.7 | 27.7 | 47.7 | 29.3 | 35.4 | 71.1 | 39.3 | 54.0 | 73.1 | 46.0 |
| 10000 | 30.0 | 54.3 | 28.1 | 50.9 | 30.0 | 33.6 | 71.2 | 40.6 | 52.0 | 74.2 | 46.5 |
| 12500 | 29.3 | 56.7 | 27.5 | 52.3 | 30.9 | 33.8 | 72.8 | 39.9 | 52.5 | 77.5 | 47.3 |
| 15000 | 29.6 | 55.9 | 28.3 | 53.9 | 30.6 | 35.0 | 72.9 | 41.0 | 53.8 | 75.8 | 47.7 |
| 17500 | 30.6 | 55.5 | 28.7 | 53.3 | 31.2 | 34.2 | 73.6 | 40.4 | 53.4 | 76.7 | 47.8 |
| 20000 | 30.0 | 57.6 | 28.3 | 54.9 | 31.1 | 37.2 | 74.6 | 40.7 | 53.6 | 79.4 | 48.7 |
| 22500 | 30.1 | 56.7 | 28.6 | 55.2 | 31.4 | 37.2 | 73.8 | 41.6 | 53.3 | 77.7 | 48.6 |
| 25000 | 31.1 | 56.0 | 28.4 | 55.2 | 31.1 | 36.2 | 74.0 | 41.0 | 54.1 | 76.8 | 48.4 |

Table 19: Full evaluation results on scaling pre-training to about 50B tokens on FineWeb.

| Train Steps | ARC-C | ARC-E | CSQA | HellaSwag | MMLU | OBQA | PiQA | SIQA | WinoG | SciQ | AVG |
|---|---|---|---|---|---|---|---|---|---|---|---|
| | | | | TLM-M trained on FineWeb raw data. | | | | | | | |
| 2500 | 22.9 | 41.2 | 28.9 | 34.3 | 26.1 | 27.6 | 64.8 | 39.3 | 52.1 | 62.8 | 40.0 |
| 5000 | 25.5 | 44.5 | 30.4 | 39.8 | 26.9 | 32.0 | 68.4 | 39.2 | 52.1 | 67.2 | 42.6 |
| 7500 | 26.8 | 45.6 | 31.4 | 44.1 | 27.6 | 30.2 | 70.9 | 38.8 | 52.2 | 70.3 | 43.8 |
| 10000 | 27.2 | 46.2 | 31.3 | 47.2 | 28.3 | 31.6 | 72.1 | 38.8 | 53.4 | 69.0 | 44.5 |
| 12500 | 26.4 | 49.2 | 32.1 | 48.7 | 28.7 | 31.6 | 71.5 | 40.1 | 52.6 | 74.7 | 45.6 |
| 15000 | 27.1 | 49.6 | 32.8 | 49.5 | 28.9 | 31.0 | 72.7 | 39.0 | 52.3 | 77.1 | 46.0 |
| 17500 | 26.4 | 50.9 | 33.8 | 51.3 | 29.3 | 31.0 | 71.9 | 39.3 | 53.0 | 78.0 | 46.5 |
| 20000 | 27.1 | 53.1 | 33.2 | 51.2 | 29.6 | 32.2 | 73.4 | 39.7 | 52.3 | 76.3 | 46.8 |
| 22500 | 27.1 | 51.2 | 34.9 | 51.7 | 29.5 | 33.4 | 73.7 | 40.1 | 52.4 | 78.0 | 47.2 |
| 25000 | 28.5 | 52.6 | 33.9 | 53.2 | 29.8 | 32.6 | 72.9 | 40.2 | 53.0 | 77.1 | 47.4 |
| | | | | TLM-M trained on PROX refined FineWeb data. | | | | | | | |
| 2500 | 25.8 | 46.8 | 27.4 | 36.1 | 27.7 | 28.8 | 63.9 | 39.3 | 51.9 | 69.1 | 41.7 |
| 5000 | 28.5 | 52.1 | 28.8 | 43.5 | 29.3 | 32.6 | 66.4 | 38.7 | 51.2 | 71.3 | 44.2 |
| 7500 | 28.2 | 52.0 | 30.6 | 45.9 | 29.9 | 33.0 | 69.3 | 39.5 | 51.7 | 71.8 | 45.2 |
| 10000 | 29.3 | 54.3 | 30.6 | 48.5 | 30.8 | 33.2 | 69.7 | 40.7 | 50.6 | 74.4 | 46.2 |
| 12500 | 28.7 | 57.8 | 30.7 | 48.1 | 31.1 | 32.6 | 72.0 | 40.4 | 52.7 | 77.4 | 47.2 |
| 15000 | 31.1 | 59.6 | 31.9 | 50.4 | 31.8 | 34.4 | 71.9 | 40.5 | 50.8 | 78.0 | 48.0 |
| 17500 | 32.6 | 60.9 | 31.9 | 51.5 | 32.2 | 33.8 | 72.3 | 39.7 | 52.5 | 78.9 | 48.6 |
| 20000 | 33.2 | 62.5 | 32.5 | 51.6 | 32.4 | 34.6 | 72.4 | 39.7 | 51.7 | 80.7 | 49.1 |
| 22500 | 34.7 | 63.6 | 32.9 | 53.3 | 32.9 | 34.8 | 73.1 | 40.3 | 54.2 | 80.5 | 50.0 |
| 25000 | 34.4 | 63.9 | 32.6 | 53.0 | 33.1 | 34.4 | 73.1 | 39.3 | 52.7 | 81.5 | 49.8 |

Table 20: Detailed evaluation results of existing base models trained on different corpora and trained using different techniques.

| ARC-C | ARC-E | CSQA | HellaSwag | MMLU | OBQA | PiQA | SIQA | WinoG | SciQ | AVG |
|---|---|---|---|---|---|---|---|---|---|---|
| | | | | TINYLLAMA-1.1B (trained on 3T tokens) | | | | | | |
| 31.5 | 59.0 | 35.5 | 57.8 | 32.8 | 33.4 | 72.8 | 40.0 | 56.0 | 82.4 | 50.1 |
| | | | | OLMO-1B (trained on 2T tokens) | | | | | | |
| 31.4 | 59.7 | 38.9 | 61.9 | 32.2 | 38.4 | 76.1 | 41.5 | 53.9 | 78.8 | 51.3 |
| | | | | PYTHIA-1.4B | | | | | | |
| 28.7 | 56.9 | 34.7 | 51.7 | 31.5 | 36.0 | 71.8 | 40.8 | 55.1 | 79.3 | 48.7 |
| | | | | PYTHIA-2.8B | | | | | | |
| 32.9 | 61.0 | 36.5 | 60.4 | 33.3 | 35.0 | 73.5 | 41.1 | 57.0 | 83.1 | 51.4 |
| | | | | SHEAREDLLAMA-1.3B (pruned from LLAMA-2-7B) | | | | | | |
| 22.4 | 39.7 | 29.3 | 36.0 | 26.4 | 28.4 | 62.6 | 39.9 | 52.0 | 71.4 | 40.8 |
| | | | | SHEAREDLLAMA-1.3B (pruned from LLAMA-2-7B, and further trained on 50B tokens) | | | | | | |
| 29.0 | 58.3 | 34.8 | 59.6 | 32.0 | 35.0 | 74.6 | 41.0 | 56.3 | 82.3 | 50.3 |
| | | | | INSTRUCTLM-1.3B (LLM data synthesis) | | | | | | |
| 28.1 | 57.9 | 32.5 | 52.3 | 30.0 | 34.0 | 74.5 | 39.9 | 56.1 | 86.9 | 49.2 |
| | | | | COSMO-1.8B (LLM data synthesis) | | | | | | |
| 33.4 | 57.0 | 31.2 | 55.1 | 32.4 | 35.2 | 71.4 | 42.0 | 54.7 | 84.4 | 49.7 |

## E.4. Evaluation Results of Continual Pre-training in Sec. 3.4

We provide full ablation results for each base model, as shown in Table 21. We can observe that PROX-D+C consistently improves average performance over PROX-D across various base models. Although the performance gain from PROX-D+C compared to PROX-D is less pronounced than the improvement of PROX-D over continual pre-training on raw OpenWebMath, this is both understandable and expected. PROX-D+C does not significantly reduce the token count beyond the reductions achieved by PROX-D alone. Given the scale of the OpenWebMath corpus, a more aggressive token removal strategy could potentially diminish the diversity of unique tokens below the threshold necessary for robust pre-training. This observation underscores the delicate balance between data refinement and maintaining sufficient linguistic variety for effective language model training, particularly when working with limited-scale corpora.

Table 21: Full ablation results on OpenWebMath Continual Pre-training (CPT). All models are tested using few-shot CoT prompts. LLEMMA and INTERNLM2-MATH are continual pre-trained models from CODELLAMA (Rozière et al., 2023) and INTERNLM2 (Team, 2023) with public available data, respectively. DEEPSEEK-LLM denotes an internal DeepSeek model, and the model trained on OpenWebMath introduced by Shao et al. (2024). Note that the unique tokens and training tokens in the column refer exclusively to the token numbers from math-specific corpora (calculated by corresponding tokenizers). [†]: MQA evaluation of INTERNLM2-BASE is based on an alternative prompt due to non-prediction issues with the original prompt. The **bolded** entries represent the best results within the same base model and CPT experiments.

| Model | Size | Method | Uniq Toks | Train Toks | GSM8K | MATH | SVAMP | ASDiv | MAWPS | TAB | MQA | MMLU STEM | SAT MATH | AVG |
|---|---|---|---|---|---|---|---|---|---|---|---|---|---|---|
| | | | | | *Existing Continual Pre-training for Reference* | | | | | | | | | |
| DEEPSEEK-LLM | 1.3B | - | - | - | 2.9 | 3.0 | - | - | - | - | - | 19.5 | 15.6 | - |
| | 1.3B | - | 14B | 150B | 11.5 | 8.9 | - | - | - | - | - | 29.6 | 31.3 | - |
| CODELLAMA (Base) | 7B | - | - | - | 11.8 | 5.0 | 44.2 | 50.7 | 62.6 | 30.6 | 14.3 | 20.4 | 21.9 | 29.1 |
| | 34B | - | - | - | 31.8 | 10.8 | 61.9 | 66.0 | 83.4 | 51.6 | 23.7 | 43.0 | 53.1 | 47.3 |
| LLEMMA | 7B | - | 55B | 200B | 38.8 | 17.2 | 56.1 | 69.1 | 82.4 | 48.7 | 41.0 | 45.4 | 59.4 | 50.9 (+21.8) |
| | 34B | - | 55B | 50B | 54.2 | 23.0 | 67.9 | 75.7 | 90.1 | 57.9 | 49.8 | 54.7 | 68.8 | 60.1 (+12.8) |
| INTERNLM2-BASE | 7B | - | - | - | 27.0 | 6.6 | 49.0 | 59.3 | 74.8 | 40.1 | 20.9[†] | 19.0 | 28.1 | 36.1 |
| | 20B | - | - | - | 50.6 | 18.8 | 72.5 | 75.9 | 93.9 | 45.4 | 33.1 | 53.7 | 59.4 | 55.9 |
| INTERNLM2-MATH | 7B | - | 31B | 125B | 41.8 | 14.4 | 61.6 | 66.8 | 83.7 | 50.0 | 57.3 | 24.8 | 37.5 | 48.7 (+12.6) |
| | 20B | - | 120B | 500B | 65.4 | 30.0 | 75.7 | 79.3 | 94.0 | 50.9 | 38.5 | 53.1 | 71.9 | 62.1 (+6.2) |
| | | | | | *Applying Data Refinement Approaches* | | | | | | | | | |
| TINYLLAMA (Base) | 1.1B | - | - | - | 2.8 | 3.2 | 10.9 | 18.0 | 20.2 | 12.5 | 14.6 | 16.4 | 21.9 | 14.7 |
| TINYLLAMA (CPT) | 1.1B | - | 15B | 15B | 6.2 | 4.8 | 22.3 | 36.2 | 47.6 | 19.3 | 11.6 | 20.7 | 25.0 | 21.5 (+6.8) |
| | 1.1B | RHO | 15B | 9B[*5] | 7.1 | 5.0 | 23.5 | 41.2 | 53.8 | - | **18.0** | - | - | - |
| | 1.1B | Rule | 6.5B | 15B | 4.5 | 2.8 | 17.5 | 29.4 | 39.3 | 15.1 | 12.4 | 19.4 | 25.0 | 18.4 (+3.7) |
| | 1.1B | PROX-D | 5.4B | 15B | **9.3** | **7.4** | 23.4 | **41.9** | 55.6 | 22.1 | 14.6 | 24.1 | 25.0 | 24.8 (+10.1) |
| | 1.1B | PROX-D+C | 5B | 15B | 9.0 | 5.6 | **23.8** | **41.9** | **56.9** | **22.2** | 15.6 | **26.8** | **31.2** | **25.7 (+11.0)** |
| LLAMA-2 (Base) | 7B | - | - | - | 14.1 | 3.8 | 39.5 | 51.6 | 63.6 | 30.9 | 12.5 | 32.9 | 34.4 | 31.5 |
| LLAMA-2 (CPT) | 7B | - | 15B | 10B | 29.6 | 13.6 | 49.2 | 61.9 | 78.4 | 36.3 | 31.9 | 40.5 | 43.8 | 42.8 (+11.3) |
| | 7B | PROX-D | 5.4B | 10B | 30.3 | 16.0 | **54.2** | 63.8 | **79.5** | **37.3** | 37.2 | **44.2** | 46.9 | 45.5 (+14.0) |
| | 7B | PROX-D+C | 5B | 10B | **30.6** | **16.8** | 50.2 | 63.7 | 79.3 | **37.3** | **40.1** | 43.8 | **53.1** | **46.1 (+14.6)** |
| CODELLAMA (Base) | 7B | - | - | - | 11.8 | 5.0 | 44.2 | 50.7 | 62.6 | 30.6 | 14.3 | 20.4 | 21.9 | 29.1 |
| CODELLAMA (CPT) | 7B | - | 15B | 10B | 31.1 | 14.8 | 51.4 | 62.1 | 81.2 | 33.6 | 30.4 | 40.5 | 43.8 | 43.2 (+14.1) |
| | 7B | PROX-D | 5.4B | 10B | **38.1** | 17.0 | 54.2 | 67.0 | **83.1** | 40.9 | **39.8** | **43.7** | 50.0 | 48.2 (+19.1) |
| | 7B | PROX-D+C | 5B | 10B | 35.6 | **17.6** | **55.8** | **67.9** | 82.7 | **41.3** | 38.9 | 42.6 | **62.5** | **49.4 (+20.3)** |
| MISTRAL (Base) | 7B | - | - | - | 40.6 | 11.4 | **65.4** | 68.5 | 87.0 | **52.9** | 32.3 | 50.0 | 56.2 | 51.6 |
| MISTRAL (CPT) | 7B | - | 15B | 10B | 44.4 | 19.2 | 65.2 | 69.6 | 88.4 | 46.6 | 43.1 | 50.8 | 65.6 | 54.8 (+3.2) |
| | 7B | PROX-D | 5.5B | 10B | 47.8 | **24.8** | 63.5 | 72.4 | 88.9 | 48.3 | 48.2 | 54.1 | 62.5 | 56.4 (+4.8) |
| | 7B | PROX-D+C | 4.7B | 10B | **51.0** | 22.4 | 64.9 | **72.9** | **89.2** | 49.8 | **53.0** | 54.2 | **75.0** | **59.2 (+7.6)** |

Besides, we report the detailed dynamic evaluation results of our continual pre-training experiments on OpenWebMath:

---

[5]RHO-1 only counts the selected tokens that are used for training (loss calculation).

- Tables 22, 23, 24, and 25 present the evaluation results for TINYLLAMA-1.1B.

- Tables 26, 27, and 28 present the evaluation results for LLAMA-2.

- Tables 29, 30, 31 present the evaluation results for CODELLAMA.

- Tables 32, 33, and 34 show the evaluation results for MISTRAL-7B.

Table 22: Full evaluation results of TINYLLAMA-1.1B continual pre-training on OpenWebMath with raw data. Note that about 1B tokens are trained per 500 steps.

| Train Steps | GSM8K | MATH | SVAMP | ASDiv | MAWPS | TAB | MQA | MMLU STEM | SAT MATH | AVG |
|---|---|---|---|---|---|---|---|---|---|---|
| 0 | 2.8 | 3.2 | 10.9 | 18 | 20.2 | 12.5 | 14.6 | 16.4 | 21.9 | 14.7 |
| 500 | 1.9 | 3.4 | 16.3 | 23.9 | 30.3 | 13.9 | 10.3 | 14.8 | 18.8 | 14.8 |
| 1000 | 3.1 | 2.2 | 16.6 | 25.6 | 32.4 | 12.5 | 12.0 | 16.6 | 25.0 | 16.2 |
| 1500 | 2.7 | 3.0 | 17.6 | 28.5 | 34.5 | 13.9 | 8.7 | 14.1 | 15.6 | 15.4 |
| 2000 | 4.5 | 3.2 | 16.4 | 28.5 | 39.0 | 15.1 | 10.2 | 16.6 | 34.4 | 18.7 |
| 2500 | 4.9 | 3.4 | 19.3 | 31.0 | 39.2 | 16.0 | 12.1 | 18.6 | 9.4 | 17.1 |
| 3000 | 4.1 | 5.2 | 19.1 | 32.0 | 43.0 | 15.3 | 9.6 | 16.1 | 18.8 | 18.1 |
| 3500 | 4.9 | 3.6 | 19.7 | 31.4 | 40.4 | 18.1 | 11.3 | 19.6 | 15.6 | 18.3 |
| 4000 | 4.8 | 4.8 | 19.5 | 33.8 | 44.5 | 16.4 | 10.7 | 19.9 | 12.5 | 18.5 |
| 4500 | 5.4 | 4.8 | 20.2 | 35.0 | 45.2 | 17.9 | 12.7 | 21.0 | 18.8 | 20.1 |
| 5000 | 5.5 | 4.6 | 22.3 | 34.6 | 42.9 | 16.0 | 10.6 | 21.7 | 28.1 | 20.7 |
| 5500 | 4.9 | 5.8 | 23.6 | 35.2 | 44.0 | 20.4 | 11.0 | 21.1 | 21.9 | 20.9 |
| 6000 | 6.1 | 4.4 | 22.8 | 36.2 | 45.4 | 17.8 | 12.7 | 21.4 | 15.6 | 20.3 |
| 6500 | 6.3 | 3.6 | 23.2 | 37.3 | 48.0 | 19.7 | 10.3 | 21.0 | 18.8 | 20.9 |
| 7000 | 6.1 | 4.6 | 22.2 | 36.6 | 46.9 | 19.4 | 12.0 | 21.5 | 21.9 | 21.2 |
| 7500 | 6.2 | 4.8 | 22.3 | 36.2 | 47.6 | 19.3 | 11.6 | 20.7 | 25.0 | 21.5 |

Table 23: Full evaluation results of TINYLLAMA-1.1B continual pre-training on OpenWebMath with data after rule-based filtering. Note that about 1B tokens are trained per 500 steps.

| Train Steps | GSM8K | MATH | SVAMP | ASDiv | MAWPS | TAB | MQA | MMLU STEM | SAT MATH | AVG |
|---|---|---|---|---|---|---|---|---|---|---|
| 0 | 2.8 | 3.2 | 10.9 | 18 | 20.2 | 12.5 | 14.6 | 16.4 | 21.9 | 14.7 |
| 500 | 3.4 | 3.6 | 13.6 | 22.5 | 25.9 | 13.1 | 14.2 | 13.5 | 28.1 | 15.3 |
| 1000 | 3.0 | 2.8 | 14.1 | 22.5 | 27.8 | 11.4 | 11.0 | 16.4 | 12.5 | 13.5 |
| 1500 | 3.6 | 3.2 | 13.6 | 24.0 | 31.2 | 13.9 | 9.2 | 18.0 | 18.8 | 15.1 |
| 2000 | 3.5 | 2.4 | 15.0 | 25.1 | 33.0 | 12.5 | 10.6 | 13.9 | 15.6 | 14.6 |
| 2500 | 3.3 | 1.6 | 15.0 | 25.3 | 33.5 | 13.7 | 11.1 | 18.1 | 25.0 | 16.3 |
| 3000 | 3.5 | 3.0 | 16.4 | 25.5 | 33.4 | 14.1 | 10.2 | 18.4 | 18.8 | 15.9 |
| 3500 | 3.2 | 3.4 | 17.2 | 27.0 | 37.7 | 14.6 | 11.2 | 13.3 | 25.0 | 17.0 |
| 4000 | 3.5 | 3.6 | 15.6 | 26.2 | 36.5 | 13.4 | 12.1 | 15.9 | 18.8 | 16.2 |
| 4500 | 4.1 | 3.8 | 15.6 | 27.9 | 38.2 | 14.9 | 11.6 | 17.1 | 18.8 | 16.9 |
| 5000 | 4.2 | 3.6 | 18.6 | 28.7 | 37.7 | 14.3 | 12.7 | 17.5 | 21.9 | 17.7 |
| 5500 | 4.1 | 3.8 | 16.3 | 29.3 | 38.4 | 14.7 | 10.8 | 17.5 | 18.8 | 17.1 |
| 6000 | 4.3 | 3.6 | 16.0 | 28.7 | 39.1 | 13.5 | 12.8 | 19.5 | 21.9 | 17.7 |
| 6500 | 4.2 | 3.2 | 16.4 | 29.5 | 39.0 | 15.1 | 11.7 | 17.9 | 21.9 | 17.7 |
| 7000 | 4.0 | 4.0 | 16.2 | 29.6 | 37.9 | 16.0 | 13.8 | 17.8 | 21.9 | 17.9 |
| 7500 | 4.5 | 2.8 | 17.5 | 29.4 | 39.3 | 15.1 | 12.4 | 19.4 | 25.0 | 18.4 |

Table 24: Full evaluation results of TINYLLAMA-1.1B continual pre-training on OpenWebMath with data after PROX-D. Note that about 1B tokens are trained per 500 steps.

| Train Steps | GSM8K | MATH | SVAMP | ASDiv | MAWPS | TAB | MQA | MMLU STEM | SAT MATH | AVG |
|---|---|---|---|---|---|---|---|---|---|---|
| 0 | 2.8 | 3.2 | 10.9 | 18 | 20.2 | 12.5 | 14.6 | 16.4 | 21.9 | 14.7 |
| 500 | 3.3 | 2.8 | 17.7 | 29.0 | 38.7 | 12.4 | 9.5 | 15.7 | 15.6 | 16.1 |
| 1000 | 4.6 | 4.0 | 18.1 | 31.6 | 41.9 | 15.9 | 11.9 | 18.2 | 25.0 | 19.0 |
| 1500 | 5.2 | 5.4 | 21.1 | 32.9 | 43.1 | 15.3 | 11.1 | 20.4 | 12.5 | 18.6 |
| 2000 | 6.8 | 5.8 | 20.2 | 33.5 | 46.6 | 18.2 | 10.7 | 20.3 | 12.5 | 19.4 |
| 2500 | 7.1 | 3.8 | 20.7 | 37.0 | 48.6 | 18.3 | 12.0 | 21.4 | 18.8 | 20.9 |
| 3000 | 7.4 | 4.4 | 22.9 | 37.1 | 50.5 | 18.3 | 12.3 | 21.2 | 25.0 | 22.1 |
| 3500 | 8.8 | 4.8 | 22.8 | 39.4 | 53.3 | 19.2 | 12.0 | 22.8 | 34.4 | 24.2 |
| 4000 | 8.6 | 4.6 | 24.0 | 38.7 | 51.4 | 18.8 | 14.8 | 24.4 | 18.8 | 22.7 |
| 4500 | 8.6 | 4.2 | 24.2 | 39.2 | 53.6 | 20.4 | 13.5 | 23.9 | 18.8 | 22.9 |
| 5000 | 8.9 | 5.2 | 24.0 | 40.0 | 52.6 | 20.0 | 13.6 | 23.9 | 18.8 | 23.0 |
| 5500 | 8.0 | 6.2 | 23.2 | 41.4 | 55.0 | 22.3 | 14.3 | 24.9 | 25.0 | 24.5 |
| 6000 | 8.3 | 5.2 | 22.2 | 39.8 | 54.0 | 24.3 | 12.6 | 25.1 | 31.2 | 24.7 |
| 6500 | 9.4 | 5.6 | 24.4 | 40.2 | 54.5 | 20.3 | 13.0 | 24.9 | 31.2 | 24.8 |
| 7000 | 9.2 | 5.8 | 25.8 | 40.6 | 55.3 | 22.5 | 12.5 | 24.5 | 21.9 | 24.2 |
| 7500 | 9.3 | 7.4 | 23.4 | 41.9 | 55.6 | 22.1 | 14.6 | 24.1 | 25.0 | 24.8 |

Table 25: Full evaluation results of TINYLLAMA-1.1B continual pre-training on OpenWebMath with data after PROX-D+C. Note that about 1B tokens are trained per 500 steps.

| Train Steps | GSM8K | MATH | SVAMP | ASDiv | MAWPS | TAB | MQA | MMLU STEM | SAT MATH | AVG |
|---|---|---|---|---|---|---|---|---|---|---|
| 0 | 2.8 | 3.2 | 10.9 | 18 | 20.2 | 12.5 | 14.6 | 16.4 | 21.9 | 14.7 |
| 500 | 4.3 | 5.0 | 16.4 | 28.8 | 36.4 | 15.3 | 11.4 | 18.5 | 15.6 | 16.9 |
| 1000 | 5.5 | 3.8 | 20.5 | 34.6 | 44.6 | 15.3 | 12.1 | 19.6 | 28.1 | 20.5 |
| 1500 | 5.2 | 4.4 | 21.4 | 34.5 | 44.7 | 16.1 | 11.2 | 21.4 | 34.4 | 21.5 |
| 2000 | 6.3 | 5.4 | 20.1 | 33.7 | 46.2 | 19.4 | 10.5 | 21.2 | 12.5 | 19.5 |
| 2500 | 7.8 | 5.4 | 22.1 | 37.0 | 49.5 | 17.9 | 13.3 | 22.9 | 21.9 | 22.0 |
| 3000 | 6.4 | 3.4 | 23.0 | 38.6 | 51.1 | 18.5 | 12.6 | 24.3 | 18.8 | 21.9 |
| 3500 | 8.5 | 4.6 | 24.1 | 40.2 | 53.8 | 22.1 | 12.5 | 23.1 | 25.0 | 23.8 |
| 4000 | 8.2 | 6.0 | 24.1 | 41.0 | 52.4 | 19.8 | 10.2 | 26.1 | 31.2 | 24.3 |
| 4500 | 8.3 | 5.4 | 24.1 | 41.3 | 54.4 | 20.6 | 15.2 | 24.2 | 28.1 | 24.6 |
| 5000 | 8.5 | 7.0 | 26.0 | 40.5 | 54.9 | 21.7 | 13.9 | 25.5 | 34.4 | 25.8 |
| 5500 | 8.7 | 4.0 | 23.2 | 41.1 | 54.8 | 20.5 | 14.4 | 26.5 | 21.9 | 23.9 |
| 6000 | 8.3 | 5.0 | 24.8 | 41.3 | 54.3 | 23.2 | 14.0 | 25.3 | 25.0 | 24.6 |
| 6500 | 8.6 | 6.4 | 24.5 | 41.6 | 55.1 | 22.2 | 14.4 | 26.5 | 25.0 | 24.9 |
| 7000 | 8.9 | 6.0 | 23.4 | 40.5 | 53.4 | 22.0 | 15.8 | 27.3 | 28.1 | 25.0 |
| 7500 | 9.0 | 4.4 | 23.8 | 41.9 | 56.4 | 22.2 | 15.6 | 26.8 | 31.2 | 25.7 |

Table 26: Full evaluation results of LLAMA-2 continual pre-training on OpenWebMath with raw data. Note that about 1B tokens are trained per 1000 steps.

| Train Steps | GSM8K | MATH | SVAMP | ASDiv | MAWPS | TAB | MQA | MMLU STEM | SAT MATH | AVG |
|---|---|---|---|---|---|---|---|---|---|---|
| 0 | 14.1 | 3.8 | 39.5 | 51.6 | 63.6 | 30.9 | 12.5 | 32.9 | 34.4 | 31.5 |
| 1k | 17.2 | 3.6 | 39.1 | 50.4 | 63.0 | 30.2 | 18.9 | 31.8 | 31.2 | 31.7 |
| 2k | 19.7 | 6.0 | 43.9 | 55.5 | 68.3 | 32.9 | 19.0 | 33.0 | 37.5 | 35.1 |
| 3k | 19.6 | 8.6 | 42.9 | 56.3 | 68.4 | 32.2 | 17.4 | 34.6 | 40.6 | 35.6 |
| 4k | 21.8 | 8.8 | 44.6 | 57.3 | 72.0 | 28.9 | 23.6 | 35.8 | 40.6 | 37.0 |
| 5k | 22.6 | 10.4 | 45.9 | 57.0 | 73.5 | 31.5 | 23.9 | 39.0 | 43.8 | 38.6 |
| 6k | 24.5 | 10.0 | 44.9 | 57.6 | 73.7 | 35.5 | 25.8 | 36.1 | 43.8 | 39.1 |
| 7k | 23.3 | 10.4 | 46.5 | 59.0 | 75.3 | 32.9 | 27.7 | 39.0 | 50.0 | 40.5 |
| 8k | 29.0 | 12.4 | 46.4 | 59.7 | 77.0 | 33.1 | 30.2 | 38.8 | 50.0 | 41.8 |
| 9k | 26.1 | 12.8 | 48.8 | 59.9 | 74.3 | 35.0 | 28.3 | 39.2 | 50.0 | 41.6 |
| 10k | 29.6 | 13.6 | 49.2 | 61.9 | 78.4 | 36.3 | 31.9 | 40.5 | 43.8 | 42.8 |

Table 27: Full evaluation results of LLAMA-2 continual pre-training on OpenWebMath with **PROX-D**. Note that about 1B tokens are trained per 1000 steps.

| Train Steps | GSM8K | MATH | SVAMP | ASDiv | MAWPS | TAB | MQA | MMLU STEM | SAT MATH | AVG |
|---|---|---|---|---|---|---|---|---|---|---|
| 0 | 14.1 | 3.8 | 39.5 | 51.6 | 63.6 | 30.9 | 12.5 | 32.9 | 34.4 | 31.5 |
| 1k | 17.1 | 7.2 | 39.8 | 51.6 | 68.4 | 31.4 | 21.4 | 35.2 | 40.6 | 34.7 |
| 2k | 21.9 | 9.2 | 43.2 | 57.0 | 72.8 | 33.1 | 24.0 | 37.6 | 56.2 | 39.4 |
| 3k | 20.5 | 10.8 | 45.7 | 58.6 | 76.2 | 35.3 | 25.8 | 38.3 | 53.1 | 40.5 |
| 4k | 27.2 | 11.8 | 45.7 | 58.7 | 76.6 | 35.9 | 29.2 | 41.0 | 31.2 | 39.7 |
| 5k | 28.9 | 14.2 | 49.3 | 60.2 | 77.9 | 38.8 | 32.8 | 41.7 | 53.1 | 44.1 |
| 6k | 31.9 | 15.0 | 51.5 | 62.0 | 79.0 | 39.2 | 33.3 | 41.4 | 68.8 | 46.9 |
| 7k | 31.5 | 16.8 | 51.9 | 63.2 | 77.9 | 36.5 | 35.9 | 43.8 | 43.8 | 44.6 |
| 8k | 30.3 | 13.8 | 51.9 | 63.7 | 80.6 | 38.3 | 36.1 | 41.3 | 59.4 | 46.2 |
| 9k | 30.6 | 14.0 | 52.7 | 62.6 | 78.7 | 37.5 | 36.1 | 43.2 | 43.8 | 44.4 |
| 10k | 30.3 | 16.0 | 54.2 | 63.8 | 79.5 | 37.3 | 37.2 | 44.2 | 46.9 | 45.5 |

Table 28: Full evaluation results of LLAMA-2 continual pre-training on OpenWebMath with **PROX-D+C**. Note that about 1B tokens are trained per 1000 steps.

| Train Steps | GSM8K | MATH | SVAMP | ASDiv | MAWPS | TAB | MQA | MMLU STEM | SAT MATH | AVG |
|---|---|---|---|---|---|---|---|---|---|---|
| 0 | 14.1 | 3.8 | 39.5 | 51.6 | 63.6 | 30.9 | 12.5 | 32.9 | 34.4 | 31.5 |
| 1k | 18.8 | 6.8 | 40.1 | 54.4 | 66.1 | 29.7 | 22.9 | 35.6 | 53.1 | 36.4 |
| 2k | 23.1 | 8.6 | 45.7 | 56.5 | 72.7 | 30.7 | 25.1 | 35.6 | 46.9 | 38.3 |
| 3k | 23.4 | 11.8 | 47.9 | 59.1 | 74.6 | 30.4 | 28.2 | 38.3 | 59.4 | 41.5 |
| 4k | 25.2 | 14.2 | 49.0 | 57.8 | 72.7 | 32.8 | 33.1 | 40.7 | 40.6 | 40.7 |
| 5k | 24.4 | 13.6 | 48.0 | 58.7 | 72.1 | 28.9 | 33.0 | 40.6 | 50.0 | 41.0 |
| 6k | 29.6 | 12.8 | 46.1 | 63.4 | 75.6 | 33.7 | 31.6 | 42.8 | 53.1 | 43.2 |
| 7k | 29.9 | 13.6 | 50.5 | 61.5 | 75.2 | 36.4 | 34.5 | 41.7 | 53.1 | 44.0 |
| 8k | 30.2 | 15.8 | 50.8 | 63.7 | 77.1 | 37.7 | 36.3 | 43.4 | 43.8 | 44.3 |
| 9k | 34.0 | 15.4 | 52.1 | 62.4 | 79.3 | 35.9 | 40.2 | 44.0 | 56.2 | 46.6 |
| 10k | 30.6 | 16.8 | 50.2 | 63.7 | 79.3 | 37.3 | 40.1 | 43.8 | 53.1 | 46.1 |

Table 29: Full evaluation results of CODELLAMA-7B continual pre-training on OpenWebMath with raw data. Note that about 1B tokens are trained per 250 steps.

| Train Steps | GSM8K | MATH | SVAMP | ASDiv | MAWPS | TAB | MQA | MMLU STEM | SAT MATH | AVG |
|---|---|---|---|---|---|---|---|---|---|---|
| 0 | 11.8 | 5.0 | 44.2 | 50.7 | 62.6 | 30.6 | 14.3 | 20.4 | 21.9 | 29.1 |
| 250 | 16.7 | 8.2 | 45.2 | 52.2 | 65.3 | 33.9 | 16.0 | 28.8 | 43.8 | 34.5 |
| 500 | 18.3 | 7.8 | 43.1 | 53.9 | 69.0 | 29.3 | 15.3 | 22.5 | 37.5 | 33.0 |
| 750 | 20.2 | 8.0 | 45.2 | 54.2 | 71.9 | 29.9 | 17.1 | 31.2 | 37.5 | 35.0 |
| 1000 | 24.7 | 9.8 | 40.6 | 58.6 | 72.7 | 29.3 | 20.7 | 31.9 | 34.4 | 35.9 |
| 1250 | 24.3 | 10.4 | 44.0 | 57.5 | 74.8 | 29.2 | 21.4 | 36.1 | 50.0 | 38.6 |
| 1500 | 26.2 | 13.2 | 48.4 | 58.8 | 75.4 | 29.4 | 28.1 | 34.9 | 50.0 | 40.5 |
| 1750 | 25.5 | 11.8 | 49.1 | 58.7 | 76.6 | 32.4 | 26.7 | 37.3 | 43.8 | 40.2 |
| 2000 | 28.0 | 13.6 | 46.3 | 61.7 | 80.0 | 33.8 | 29.4 | 37.2 | 50.0 | 42.2 |
| 2250 | 27.7 | 13.6 | 48.9 | 62.2 | 80.3 | 32.5 | 28.9 | 39.1 | 59.4 | 43.6 |
| 2500 | 31.1 | 14.8 | 51.4 | 62.1 | 81.2 | 33.6 | 30.4 | 40.5 | 43.8 | 43.2 |

Table 30: Full evaluation results of CODELLAMA continual pre-training on OpenWebMath with **PROX-D**. Note that about 1B tokens are trained per 250 steps.

| Train Steps | GSM8K | MATH | SVAMP | ASDiv | MAWPS | TAB | MQA | MMLU STEM | SAT MATH | AVG |
|---|---|---|---|---|---|---|---|---|---|---|
| 0 | 11.8 | 5.0 | 44.2 | 50.7 | 62.6 | 30.6 | 14.3 | 20.4 | 21.9 | 29.1 |
| 250 | 21.1 | 9.2 | 48.7 | 56.1 | 71.3 | 33.4 | 22.2 | 34.1 | 50.0 | 38.5 |
| 500 | 23.7 | 11.6 | 49.8 | 57.4 | 74.7 | 32.9 | 28.5 | 35.8 | 59.4 | 41.5 |
| 750 | 25.1 | 15.4 | 48.1 | 58.9 | 78.8 | 36.8 | 29.4 | 37.6 | 53.1 | 42.6 |
| 1000 | 28.4 | 14.2 | 50.9 | 61.2 | 79.8 | 36.7 | 27.7 | 37.6 | 50.0 | 42.9 |
| 1250 | 33.0 | 15.2 | 49.3 | 62.9 | 81.1 | 33.4 | 32.8 | 41.0 | 46.9 | 44.0 |
| 1500 | 36.0 | 15.0 | 54.2 | 65.0 | 81.0 | 39.3 | 34.1 | 42.0 | 62.5 | 47.7 |
| 1750 | 34.7 | 14.6 | 53.1 | 63.6 | 83.3 | 40.6 | 35.9 | 43.4 | 62.5 | 48.0 |
| 2000 | 35.7 | 17.6 | 53.3 | 65.4 | 83.5 | 42.4 | 37.1 | 42.4 | 56.2 | 48.2 |
| 2250 | 37.2 | 18.8 | 54.5 | 65.4 | 83.2 | 41.9 | 41.0 | 44.9 | 71.9 | 51.0 |
| 2500 | 38.1 | 17.0 | 54.2 | 67.0 | 83.1 | 40.9 | 39.8 | 43.7 | 50.0 | 48.2 |

Table 31: Full evaluation results of CODELLAMA continual pre-training on OpenWebMath with **PROX-D+C**. Note that about 1B tokens are trained per 250 steps.

| Train Steps | GSM8K | MATH | SVAMP | ASDiv | MAWPS | TAB | MQA | MMLU STEM | SAT MATH | AVG |
|---|---|---|---|---|---|---|---|---|---|---|
| 0 | 11.8 | 5.0 | 44.2 | 50.7 | 62.6 | 30.6 | 14.3 | 20.4 | 21.9 | 29.1 |
| 250 | 18.1 | 10.2 | 46.0 | 54.5 | 71.9 | 33.0 | 21.3 | 34.4 | 50.0 | 37.7 |
| 500 | 22.4 | 10.0 | 50.3 | 59.7 | 76.4 | 31.3 | 26.1 | 36.0 | 59.4 | 41.3 |
| 750 | 26.8 | 11.4 | 51.2 | 61.0 | 78.5 | 34.9 | 26.4 | 38.0 | 53.1 | 42.4 |
| 1000 | 29.0 | 14.4 | 54.1 | 62.8 | 80.1 | 36.9 | 34.2 | 40.4 | 62.5 | 46.0 |
| 1250 | 31.4 | 15.0 | 51.7 | 63.8 | 81.1 | 37.2 | 32.5 | 41.4 | 75.0 | 47.7 |
| 1500 | 31.5 | 17.4 | 53.4 | 64.4 | 80.7 | 39.6 | 35.4 | 41.6 | 71.9 | 48.4 |
| 1750 | 33.7 | 15.2 | 50.6 | 64.3 | 81.5 | 39.2 | 36.1 | 40.5 | 53.1 | 46.0 |
| 2000 | 36.2 | 16.0 | 54.7 | 65.1 | 83.1 | 39.9 | 39.1 | 43.4 | 71.9 | 49.9 |
| 2250 | 37.1 | 16.6 | 55.3 | 65.6 | 82.4 | 41.3 | 36.5 | 42.7 | 75.0 | 50.3 |
| 2500 | 35.6 | 17.6 | 55.8 | 67.9 | 82.7 | 41.3 | 38.9 | 42.6 | 62.5 | 49.4 |

Table 32: Full evaluation results of MISTRAL-7B continual pre-training on OpenWebMath with raw data. Note that about 1B tokens are trained per 1000 steps.

| Train Steps | GSM8K | MATH | SVAMP | ASDiv | MAWPS | TAB | MQA | MMLU STEM | SAT MATH | AVG |
|---|---|---|---|---|---|---|---|---|---|---|
| 0 | 40.6 | 11.4 | 65.4 | 68.5 | 87.0 | 52.9 | 32.3 | 50.0 | 56.2 | 51.6 |
| 1k | 31.6 | 12.0 | 56.5 | 66.0 | 80.1 | 43.9 | 27.1 | 45.1 | 56.2 | 46.5 |
| 2k | 32.4 | 10.8 | 54.7 | 63.5 | 82.6 | 40.8 | 31.6 | 45.7 | 59.4 | 46.8 |
| 3k | 33.6 | 14.8 | 60.4 | 64.7 | 84.5 | 43.5 | 33.1 | 47.2 | 68.8 | 50.1 |
| 4k | 35.1 | 14.8 | 58.7 | 65.2 | 84.4 | 41.2 | 38.5 | 47.3 | 62.5 | 49.7 |
| 5k | 33.4 | 16.0 | 59.3 | 65.0 | 83.8 | 46.7 | 34.6 | 49.1 | 62.5 | 50.0 |
| 6k | 38.7 | 16.6 | 61.5 | 68.1 | 86.1 | 47.4 | 35.3 | 48.5 | 37.5 | 48.9 |
| 7k | 39.6 | 17.2 | 60.5 | 68.2 | 86.2 | 44.4 | 38.5 | 49.3 | 53.1 | 50.8 |
| 8k | 44.0 | 16.4 | 64.5 | 69.8 | 88.7 | 45.5 | 41.3 | 50.6 | 59.4 | 53.4 |
| 9k | 43.9 | 19.4 | 63.7 | 69.7 | 87.6 | 44.9 | 42.9 | 51.0 | 62.5 | 54.0 |
| 10k | 44.4 | 19.2 | 65.2 | 69.6 | 88.4 | 46.6 | 43.1 | 50.8 | 65.6 | 54.8 |

Table 33: Full evaluation results of MISTRAL-7B continual pre-training on OpenWebMath with **PROX-D**. Note that about 1B tokens are trained per 1000 steps.

| Train Steps | GSM8K | MATH | SVAMP | ASDiv | MAWPS | TAB | MQA | MMLU STEM | SAT MATH | AVG |
|---|---|---|---|---|---|---|---|---|---|---|
| 0 | 40.6 | 11.4 | 65.4 | 68.5 | 87.0 | 52.9 | 32.3 | 50.0 | 56.2 | 51.6 |
| 1k | 36.8 | 14.6 | 57.2 | 66.1 | 83.1 | 45.7 | 32.6 | 47.7 | 59.4 | 49.2 |
| 2k | 38.5 | 17.0 | 57.9 | 69.0 | 86.3 | 44.7 | 33.6 | 49.2 | 56.2 | 50.3 |
| 3k | 40.0 | 19.0 | 59.3 | 68.7 | 87.0 | 46.8 | 41.0 | 48.0 | 68.8 | 53.2 |
| 4k | 38.5 | 20.4 | 59.3 | 66.2 | 85.1 | 42.6 | 42.8 | 49.5 | 68.8 | 52.6 |
| 5k | 42.5 | 20.2 | 63.0 | 70.5 | 86.6 | 47.2 | 43.4 | 49.8 | 62.5 | 54.0 |
| 6k | 46.8 | 17.8 | 62.5 | 72.7 | 88.2 | 51.2 | 47.7 | 51.3 | 56.2 | 54.9 |
| 7k | 47.5 | 22.4 | 64.1 | 71.8 | 89.1 | 51.4 | 47.9 | 52.4 | 65.6 | 56.9 |
| 8k | 44.6 | 23.8 | 63.2 | 70.8 | 87.7 | 47.6 | 49.1 | 54.1 | 65.6 | 56.3 |
| 9k | 46.6 | 24.6 | 61.6 | 72.3 | 86.4 | 46.9 | 49.8 | 53.2 | 65.6 | 56.3 |
| 10k | 46.7 | 22.6 | 63.5 | 72.4 | 88.9 | 48.3 | 48.2 | 54.1 | 62.5 | 56.4 |

Table 34: Full evaluation results of Mistral-7B continual pre-training on OpenWebMath with **PROX-D+C**. Note that about 1B tokens are trained per 1000 steps.

| Train Steps | GSM8K | MATH | SVAMP | ASDiv | MAWPS | TAB | MQA | MMLU STEM | SAT MATH | AVG |
|---|---|---|---|---|---|---|---|---|---|---|
| 0 | 40.6 | 11.4 | 65.4 | 68.5 | 87.0 | 52.9 | 32.3 | 50.0 | 56.2 | 51.6 |
| 1k | 30.9 | 16.0 | 60.1 | 64.5 | 85.3 | 40.8 | 33.9 | 48.0 | 59.4 | 48.8 |
| 2k | 40.3 | 17.6 | 63.0 | 66.3 | 86.2 | 48.0 | 33.9 | 48.7 | 53.1 | 50.8 |
| 3k | 42.4 | 17.8 | 59.6 | 69.1 | 85.7 | 50.1 | 38.5 | 49.9 | 59.4 | 52.5 |
| 4k | 43.8 | 20.4 | 63.7 | 69.3 | 88.2 | 46.2 | 46.3 | 50.9 | 65.6 | 54.9 |
| 5k | 42.5 | 18.4 | 59.3 | 69.6 | 87.9 | 44.3 | 46.1 | 51.9 | 65.6 | 54.0 |
| 6k | 47.7 | 21.8 | 62.7 | 71.7 | 89.2 | 47.9 | 48.4 | 54.0 | 68.8 | 56.9 |
| 7k | 46.8 | 21.6 | 62.9 | 72.1 | 88.4 | 50.1 | 46.4 | 52.5 | 68.8 | 56.6 |
| 8k | 48.4 | 21.6 | 65.0 | 72.7 | 89.2 | 51.1 | 49.4 | 52.9 | 65.6 | 57.3 |
| 9k | 48.5 | 24.8 | 64.4 | 72.6 | 88.3 | 50.7 | 48.1 | 53.4 | 62.5 | 57.0 |
| 10k | 51.0 | 22.4 | 64.9 | 72.9 | 89.2 | 49.8 | 53.0 | 54.2 | 75.0 | 59.2 |

# F. Analysis

## F.1. Token Length Distribution

Table 35: Average length of token per document for different refining methods.

| Methods | General Domain | Math Domain |
|---|---|---|
| N/A | 1217.5 | 1815.8 |
| Rule | 1329.4 | 1955.6 |
| PROX (ours) | 2004.8 | 1734.9 |

As previously discussed in §4, our analysis reveals a notable document length distribution shift in the data refined by PROX, specifically a significant increase in the average token length (from 1217.5 to 2004.8 tokens per document). When further compared to the rule-based method (we compare to FineWeb rules), we only observe a marginal increase in token length within the general domain (from 1217.5 to 1329.4 tokens).

Interestingly, in the math domain, we observe an opposite trend. The raw data shows an average token length of 1815.8, which our method reduces to 1734.9, while the rule-based method increases it to 1955.6. And the training performance in Table 5 follows the order: PROX > original > rule-based method for TINYLLAMA-1.1B. This again implies that mathematical documents used for pre-training exhibit significant differences in distribution and characteristics compared to those in the general domain.

## F.2. Case Studies

We provide several cases to qualitatively illustrate the refinement effect of PROX, as shown in Tables 36-37. For the general domain, using RedPajama-V2 as an example, we observe that PROX can drop low-information documents, remove meaningless content such as navigation bars, and replace URL links (see Table 36). In the mathematics domain, PROX demonstrates the ability to eliminate documents with minimal relevance to mathematical reasoning and remove less important elements like functional buttons (see Table 37). These refinements enhance the quality and relevance of the processed data across different domains.

## F.3. Error Analysis

As shown in Table 38, the failure ratio across both refining stages (document-level and chunk-level) and domains (General and Math) is remarkably low ($< 0.5\%$). This demonstrates that ProX's refining tasks are well-suited for small models. Specifically, for the General domain, failure ratios are $0.04\%$ for document-level and $0.36\%$ for chunk-level refining, with an average of 3.7 function calls per program in the chunk-level stage. For the Math domain, these ratios are $0.06\%$ and $0.11\%$, respectively, with an average complexity of 2.7 function calls at the chunk-level stage.

Despite the low failure rates, we observed two prevalent failure cases in ProX's programs:

1. **Repeated output or empty output:** This occurs when a program inadvertently generates duplicate outputs or fails to produce any meaningful results. Such failures are typically linked to improper loop conditions or insufficient constraints in processing logic.

2. **Non-existent target removal:** In some cases, ProX's programs attempt to remove a string or line that does not exist in the input data. This leads to incomplete execution or errors in the program output, particularly in datasets with irregular formats or unexpected variations.

As shown in Table 39, we present two failure cases to illustrate instances of repeated output and non-existent target strings.

Table 36: Cases from RedPajama-V2 after applying PROX. Text in red indicates content to be removed or replaced. "..." denotes omitted content due to limited space.

| Case 1 |
|---|

TagCollegeEducationJournalismWar

: Michael Lewis

ContributorMichael Lewis

Michael Lewis is possibly the most entertaining nonfiction writer alive. If that's not true it's at least close to true. Liar's Poker, Moneyball, The Blind Side, his NYT article about Jonathan Lebed (Google it): what's not to love?

504: How I Got Into College

Act Two: My Ames is True

Writer Michael Lewis tells the story of a man named Emir Kamenica, whose path to college started with fleeing the war in Bosnia and becoming a refugee in the United States. Then he had a stroke of luck: a student teacher read an essay he'd plagiarized from a book he'd stolen from a library back in Bosnia, and was so impressed that she got him out of a bad high school and into a much better one.

Act Three

Michael Lewis' story continues, and he figures out why Emir Kamenica insists on remembering, and telling, the story of his life the way he does — even when he finds out that some of the facts may be wrong.

**Output by PROX:**
```
drop_doc()
```

| Case 2 |
|---|

Home ¿ Staff ¿ Staff search ¿ Dr Tim Overton
Dr Tim Overton BSc PhD
School of Chemical EngineeringSenior Lecturer
Telephone (+44) (0) 121 414 5306Emailt.w.overton@bham.ac.uk
AddressSchool of Chemical EngineeringUniversity of Birmingham
B15 2TT
Dr Tim Overton is a biochemist and molecular microbiologist who is interested in applying molecular biology and single-cell techniques to understand and develop bioprocesses. He is active in microbial flow cytometry research and collaborates widely with bioprocess engineers, molecular microbiologists, cell biologists and environmental microbiologists to develop new methods of answering fundamental questions on a single-cell level.
His research also focuses on using bacteria to make useful products such as protein drugs and small molecules, and the bacterial responses to stress encountered in such processes. Current and recent research funding has come from the BBSRC, TSB and EU FP7. He is the director of the MSc in Biochemical Engineering. Pages: 1 3 4

...

Google scholar: http://scholar.google.co.uk/citations?user=tF_eBKEAAAAJ
...

**Output by PROX:**
```
keep_doc()
remove_lines(line_start=0, line_end=5)
normalize(source_str="http://scholar.google.co.uk/citations?user",
target_str="")
normalize(source_str="Pages:  1 3 4", target_str="")
 ...
```

Table 37: Cases from OpenWebMath after applying PROX. Text in red indicates content to be removed or replaced. "..." denotes omitted content due to limited space.

| Case 1 |
|---|

## unhybridized pi bonds

$sp, sp^2, sp^3, dsp^3, d^2sp^3$

Tatiana 4B

Posts: 30

Joined: Fri Sep 28, 2018 12:28 am

### unhybridized pi bonds

...

### Re: unhybridized pi bonds

I am not too sure in my knowledge about this, but I think that both have hybridized orbitals. Since hybridization is defined as the phenomenon of intermixing of the orbitals such as sp, sigma and pi bonds are just different types of covalent bonds formed depending on the way the atomic orbitals hybridize with each other. Sigma bonds are a result of when the overlap of orbitals of two atoms takes place along the line joining the two orbitals, while pi bonds are when two atoms overlap due to the sideways overlap of their 'p' orbitals.

Hannah Yates 1K

Posts: 59

Joined: Fri Sep 28, 2018 12:27 am

### Re: unhybridized pi bonds

I am also not too sure on my answer, but I am pretty sure that a sigma bond has just hybridized orbitals, but the reason a pi bond can form is because of an extra (not hybridized) p orbital. This allows for a double and triple bond to form.

**Output by PROX:**
```
drop_doc()
```

| Case 2 |
|---|

Solution - Trigonometric Identities

Account
Register

Share

Books Shortlist
ConceptTrigonometric Identities

Question

Prove the following trigonometric identities:

(i) $\frac{\sin\theta}{1-\cos\theta} = \mathrm{cosec}\theta + \cot\theta$
Solution

You need to to view the solution
Is there an error in this question or solution?

Reference Material

Solution for concept: Trigonometric Identities. For the course CBSE
S

**Output by PROX:**
```
keep_doc()
remove_lines(line_start=0, line_end=7)
remove_lines(line_start=18, line_end=24)
```

Table 38: Failure ratio and average complexity (function calls) for ProX refining stages across domains.

| Domain | Failure Ratio (doc-level) | Failure Ratio (chunk-level) | Complexity (AVG. function calls) |
|---|---|---|---|
| General Domain | 0.04% | 0.36% | 3.7 |
| Math Domain | 0.06% | 0.11% | 2.7 |

Table 39: Failure cases from RedPajama-V2 during applying PROX. "..." denotes omitted content due to limited space. The notation like `[004]` is used to indicate the line number.

| Case 1: Repeated output (or Empty output) |
|---|

...

`[004]` P: 114 1. The problem statement, all variables and given/known data Mercury is poured into a U-tube as in Figure P15.18a....Basically I don't understand why you would know to set the two volumes equal to each other? How do you know the volumes are the same?

...

`[007]` Related Discussions Mechanical Engineering 6 Introductory Physics Homework 0 General Engineering 1 Introductory Physics Homework 2 Introductory Physics Homework 2

**Output by PROX:**
```
remove_lines(start=1, end=1)
remove_lines(start=6, end=6)
remove_lines(start=7, end=7)
remove_lines(start=7, end=7)
remove_lines(start=7, end=7)
remove_lines(start=7, end
```

| Case 2: Non-existent target string |
|---|

...

`[195]` 18. Sathyamoorthi, C. R., Mbekomize, C., Mapharing, M., & Selinkie, P. (2018). The Impact of Corporate Governance on Working Capital Management Efficiency: Evidence from the Listed Companies in the Consumer Services Sector in Botswana. International Journal of Economics and Finance, 10, 135. https://doi.org/10.5539/ijef.v10n12p135

`[196]` 19. Vu, T. M. T., Tran, C. Q., Doan, D. T., & Le, T. N. (2020). Determinants of Capital Structure: The Case in Vietnam. Journal of Asian Finance, Economics, And Business, 7(9), 159-168. https://doi.org/10.13106/jafeb.2020.vol7.no9.159

...

**Output by PROX:**
```
# Analysis: this 'source_str' can not be found in the original text
normalize(source_str="https://doi.org/10.13106/jafeb.2020.vol6.no2.53",
target_str="")
```

### F.4. Computing Overhead Analysis

According to Kaplan et al. (2020), both training and inference computational FLOPs for Transformer-based Language Models (denoted as $C_{\text{train}}$ and $C_{\text{inference}}$) can be approximated as the product of model parameters (non-embedding parameter) $N$ and the number of tokens $D$. This can be expressed as:

$$C_{\text{train}} \approx 6 \cdot N D_{\text{train}}, \tag{9}$$

$$C_{\text{inference}} \approx 2 \cdot N \left( D_{\text{prefill}} + D_{\text{decode}} \right). \tag{10}$$

In PROX, we go through two data refining stages before final training, which incurs additional inference-time computational FLOPs. Suppose the refining model parameter for each stage is denoted as $N_{\text{refine}}$, and the raw data size in tokens is $D_{\text{raw}}$.

For the first document-level stage, the computational cost can be approximated as:

$$C_{\text{doc}} \approx 2 \cdot N_{\text{refine}} \left( D_{\text{raw}} + D_{\text{output}} \right) \approx 2 \cdot N_{\text{refine}} D_{\text{raw}}, \quad (\text{suppose } D_{\text{output}} \ll D_{\text{raw}}) \tag{11}$$

resulting in a new pool of data sized $D_{\text{doc}}$.

Similarly, for the second chunk-level stage, the computational cost is:

$$C_{\text{chunk}} \approx 2 \cdot N_{\text{r}} \left( D_{\text{doc}} + D_{\text{output}} \right) \approx 2 \cdot N_{\text{r}} D_{\text{doc}}, \quad (\text{suppose } D_{\text{output}} \ll D_{\text{doc}}) \tag{12}$$

which produces the final refined data size of $D_{\text{PROX}}$.

Thus, the total computational overhead for PROX can be calculated as the sum of the two stages:

$$C_{\text{PROX}} = C_{\text{doc}} + C_{\text{chunk}} \approx 2 \cdot N_{\text{doc\_refine}} D_{\text{raw}} + 2 \cdot N_{\text{chunk\_refine}} D_{\text{doc}}. \tag{13}$$

In general, we use refining models with the same sizes, so the final inference overhead can be estimated as

$$C_{\text{PROX}} \approx 2 \cdot N_{\text{refine}} (D_{\text{raw}} + D_{\text{doc}}). \tag{14}$$

Additionally, we omit the FLOPs for fine-tuning since they are negligible compared to the large-scale pre-training and inference FLOPs.

