# OpenReview forum: "Programming Every Example: Lifting Pre-training Data Quality Like Experts at Scale"
_ICML.cc/2025/Conference — ICML 2025 poster_

### Official Review · Reviewer_T2Yk · 2025-03-13

**Overall Recommendation:** 3

**Summary:**

This work proposes a new method to clean data for LM pre-training. Their method is based on a small model, which can use/combine/create functions/programs to clear and transform documents (getting rid of noise and unnecessary things). Empirically, they observe a performance improvement in downstream tasks.

**Claims And Evidence:**

The claims made are supported by clear and convincing evidence.

**Essential References Not Discussed:**

NA

**Experimental Designs Or Analyses:**

The experiments and analyses are sound.

**Methods And Evaluation Criteria:**

The method makes sense and is novel: using a small LM to clean/modify data, as far as I know, is a novel approach. The authors evaluate their method quite extensively.

**Other Comments Or Suggestions:**

NA

**Other Strengths And Weaknesses:**

NA

**Questions For Authors:**

- In Figure 7, what is the main reason why documents get lengthier after cleaning? I would expect the opposite (maybe shorter documents are getting deleted?).
- In the first plot of Figure 7, shouldn't the two areas under the curves be equal, one? One of them seems much greater than the other... maybe you're plotting the frequency and not density?

**Relation To Broader Scientific Literature:**

The relation seems clear from their related work section and, from what I get, their method offers novelty in how they use a LM to clean/modify lower quality documents.

**Theoretical Claims:**

NA

---

> ### Author Rebuttal · Authors · 2025-03-31
>
> Thank you for recognizing our extensive experiments, method soundness and novelty! We are happy to answer all your questions.
>
> $\textrm{\color{blue}Question 1}$
> ```
> What is the main reason why documents get lengthier after cleaning? I would expect the opposite (maybe shorter documents are getting deleted?)
> ```
> Yes, as you correctly pointed out, the primary reason for the longer average document length in ProX-refined data is that general text corpora—**even after extensive rule-based filtering**—still contain many **short and noisy** documents. These low-quality documents are often discarded during the document-level refining stage, leading to a curated dataset with naturally longer documents.
>
> At the chunk-level refining stage, we observe a 5–10% reduction in total tokens depending on the corpus, and document lengths correspondingly decrease. However, they **still remain consistently longer than those in the original corpus**, indicating that ProX effectively removes noise while retaining substantial and meaningful content.
>
> $\textrm{\color{blue}Question 2}$
> ```
> Maybe you're plotting the frequency and not density?
> ```
> Thank you for your careful reading! After checking it, we do not perform normalization on the Y-axis, thus showing the frequency instead of density. We will correct this typo in the updated version.

---

> > ### Comment · Reviewer_T2Yk · 2025-04-03
> >
> > Dear authors,
> >
> > Thank you for your responses! For now, I will keep my overall recommendation.

---

> > > ### Author Response · Authors · 2025-04-04
> > >
> > > Thank you! If you need any further explanations, please do not hesitate to let us know.

---

### Official Review · Reviewer_4XW9 · 2025-03-14

**Overall Recommendation:** 4

**Summary:**

The paper introduces the ProX framework which trains a model that generates short program instructions to clean pre-training documents.
The authors prompt large-language models to score the quality and format of documents and to remove various website artifacts (header, footers, URLs, and navigation elements). The generated programs unify both document filtering and chunk rewriting.

The paper includes experiments which show that the proposed method boosts the performance of language models when applied to the pre-training data. The method performs well across three model scales (up to 1B) and different pre-training datasets, and outperforms various baselines.

**Claims And Evidence:**

The central claim of superior pre-training performance is well supported by many experiments, even though the paper should compare to stronger baselines (see below). The authors also show that at larger model scale the gains in performance can justify the increased inference cost of running their ProX model on the pre-training corpus.

Another contentious issue is the comparison to methods which directly rewrite documents using LMs. The authors argue that these methods are less scalable, but it is not clear whether a strong 0.7B model might not be sufficient to rewrite data directly and the experiments do not include this relevant baseline.

**Essential References Not Discussed:**

n/a

**Experimental Designs Or Analyses:**

The choice of model sizes, data scales and evaluation tasks is appropriate for an academic paper on pre-training data.

However, I believe that document filtering and chunk-level rewriting are distinct contributions and would benefit from better individual experiments:

1. Table 2 shows how well ProX curates data from a raw corpus. However, it misses the current best curation baselines, FineWeb-Edu and DCLM-baseline. FineWeb-Edu is especially important, since document-level ProX is re-uses the FineWeb-Edu prompt (Appendix A), but using a different threshold and augmented with a format prompt.
The comparison in Table 7 in the appendix would be important to show in Table 2, and the "kept ratio" would also be important for the other methods in Table 2.
2. Table 3 shows how well ProX performs in curating a pre-processed corpus (C4). Here all baselines only perform document filtering, so showing separate rows for either document filtering, chunk rewriting, or both would be important to understand the source of the gain. Similarly, Figure 6 shows how much ProX improves the performance of strong existing corpora (in my view, the most compelling evidence for ProX). Here it would also be very useful to understand whether the performance edge is mostly due to the chunk rewriting or additional filtering.
3. The chunk-level rewriting could be compared to other approaches of rewriting text. A good baseline would training the same ProX base model to generate the cleaned document (with headers, footers, URLs, navigation removed) directly.

**Methods And Evaluation Criteria:**

The proposed method is good solution to the practical problem of selecting and rewriting pre-training data. The evaluation of the method follows standard practices from previous papers.

**Other Comments Or Suggestions:**

It would be useful to discuss deduplication as a standard part of the data curation pipeline and its relation to ProX.

**Other Strengths And Weaknesses:**

n/a

**Questions For Authors:**

Can you explain what seed data the fasttext baseline is trained on?

**Relation To Broader Scientific Literature:**

Both prompt-based selection and chunk rewriting are established techniques for curating training data.
In fact, the selection strategy is very similar to FineWeb-Edu (Penedo et al., 2024a).
The major novelity is to use structured commands (string replacement and line removal) to modify the pre-training data, which has previously been done by using LMs to rewrite the entire document. While the authors that their method is more scalable (which I am not sure about) and may be more reliable (I am tempted to agree), there is no evidence to compare between these sophisticated methods. The authors also argue that they unify chunk rewriting and document filtering in a single programming framework, but the practical value of this is not clear, especially as multiple models are still used.

Besides novelty, the released models are undoubedly useful resources to the community.

**Theoretical Claims:**

n/a

---

> ### Author Rebuttal · Authors · 2025-03-31
>
> Thanks for recognizing the practicality of our method and its strong empirical results across model scales and datasets. For your questions and suggestions:
>
> $\textrm{\color{blue}Question 1}$
> ```
> The comparison in Table7 in the appendix would be important to show in Table2, and the "kept ratio" would also be important in Table 2.
> ```
> Thank you for your careful reading and for recognizing that ProX achieves a higher kept ratio than FineWeb-Edu. We will update the presentation accordingly in our revised version.
> Regarding the kept ratio of rule-based filtering methods on **RedPajama-V2**, we summarize the results as follows. Notice that **aggressively ensembling these filtering methods does not show very obvious improvement (+0.2%) but only keeps 50% of the data**.
>
> | Filtering Method      | Kept Ratio | Avg. Performance Gain |
> |-----------------------|------------|-----------------------|
> | C4                    | 73.60%     | 0.50%                 |
> | Gopher                | 70.50%     | 0.20%                 |
> | FineWeb               | 84.50%     | 0.70%                 |
> | C4 + Gopher + FineWeb | 50.00%     | 0.20%                 |
> | **ProX-D**               | 30.00%     | **1.20%**                 |
> | **ProX-D+C**              | 25.60%     | **2.50%**                 |
>
> On FineWeb, we observe the following:
>
> | Method      | Kept Ratio | Avg. Performance |
> |-------------|------------|------------------|
> | FineWeb-Edu | 8.60%      | 45.2             |
> | **ProX**        | **28.00%**     | 45.2             |
>
> Despite keeping more than **3×** the data, ProX achieves **comparable downstream performance** to FineWeb-Edu, highlighting its effectiveness in retaining high-quality content.
>
> $\textrm{\color{blue}Question 2}$
> ```
> Here all baselines only perform document filtering, so showing separate rows for either document filtering, chunk rewriting, or both would be important to understand the source of the gain.
> ```
> In Table 3, we use only prox's doc refining performance to compare with other baseline data selection methods (please see our experiment setting from line 294 to line 296) This is mainly to show that using a very small LM on quality refinement task is indeed very simple and effective compared to other data selection baselines.
>
>
> $\textrm{\color{blue}Question 3}$
> ```
> The chunk-level rewriting could be compared to other approaches of rewriting text. A good baseline would training the same ProX base model to generate the cleaned document (with headers, footers, URLs, navigation removed) directly.
> ```
> Thank you for the insightful suggestion. While rewriting is a valid approach to improve pre-training data quality, it incurs high inference costs due to full-document generation and relies heavily on large models. Recent work like WRAP and Nemotron-CC uses models over 7B (e.g., Mistral NeMo 12B). In contrast, ProX improves data quality with much lower compute by using a small 0.3B model and concise outputs. We believe efficient rewriting would require a strong base model and leave this as future work.
>
> $\textrm{\color{blue}Suggestion}$
> ```
> Include deduplication as a standard part of the data curation pipeline and its relation to ProX.
> ```
> Thank you for the suggestion! We have discussed in the related work section that deduplication plays an important role in data preprocessing—it reduces redundancy, improves training stability, and enhances efficiency. ProX is orthogonal to deduplication, as it focuses on the quality of individual samples rather than redundancy. In practice, ProX can be applied after deduplication to further improve data quality while saving compute. We will include this discussion in the revised related work section.
>
> $\textrm{\color{blue}Question 4}$
> ```
> Can you explain what seed data the fasttext baseline is trained on?
> ```
> To ensure a fair comparison, we re-trained the fastText classifier on the exact same training data used for our ProX doc-level refining models. All documents labeled with `drop_doc()` are treated as negative samples, while those labeled with `keep_doc()` are treated as high-quality samples. We trained the language model on FastText filtered data from scratch using all the same configuration as the other experiments in Table 2.

---

> > ### Comment · Reviewer_4XW9 · 2025-04-02
> >
> > Question 1:
> >
> > Thank you! This representation of Table 1 is much better in my view. However, why not add FineWeb-Edu to it (which should also achieve a 1.2% avg performance gain if I'm not mistaken)? I also believe it is important to mention the relationship to the FineWeb-Edu prompt in the main text.
> >
> > Question 2:
> >
> > Thank you for clarifying these results for me. I had missed the note in line 294-296.
> >
> > Question 3:
> >
> > The rewriting operation in WRAP seems quite challenging (converting to different formats etc.). However, ProX performs fairly simple operations and I would not be surprised that a fine-tuned 0.3B model could regenerate the source document with headers / footers / URLs removed. Fine-tuning a model to generate commands will still be a more efficient option, but it would be good to provide evidence for the claims that the ProX framework is more robust than regenerating the document.
> >
> > Suggestion:
> >
> > Great!
> >
> > Question 4:
> >
> > This sounds like it is exactly the right baseline. Thanks!

---

> > > ### Author Response · Authors · 2025-04-04
> > >
> > > We are glad to hear that our responses helped clarify your concerns, and we truly appreciate your timely follow-up and positive acknowledgment. Thank you again for the thoughtful feedback. It's been very helpful in strengthening the paper.
> > >
> > > $\textrm{\color{blue}Further Response to Question 1}$
> > > ```
> > > However, why not add FineWeb-Edu to it (which should also achieve a 1.2% avg performance gain if I'm not mistaken)?
> > > ```
> > > Thank you for the thoughtful suggestion! We believe you are referring to Table 2 (as Table 1 focuses solely on the function design of ProX). Currently, however, all experiments reported in Table 2 are conducted on the RedPajama-V2 dataset, rather than on FineWeb. In Section 3.2 and Table 2, our primary goal is to evaluate the effectiveness of ProX in comparison to traditional rule-based filtering methods.
> > >
> > > That said, we agree that FineWeb-Edu represents a strong baseline, and we believe its classifier could also be applied to RedPajama-V2. To highlight the compatibility and added value of ProX, we include additional experiments in Figure 6, where we apply ProX refining on FineWeb-Edu data. These experiments, conducted on 1.7B models, further demonstrate that ProX can enhance data quality even when starting from a carefully filtered dataset like FineWeb-Edu.
> > >
> > > ```
> > > I also believe it is important to mention the relationship to the FineWeb-Edu prompt in the main text.
> > > ```
> > > Thank you for pointing it out. We will add this detail (currently more in Appendix A.1) in Section 2 in the next version.
> > >
> > > ---
> > >
> > > $\textrm{\color{blue}Further Response to Question 3}$
> > > ```
> > > I would not be surprised that a fine-tuned 0.3B model could regenerate the source document with headers / footers / URLs removed.
> > > ```
> > > Thank you for raising this insightful point. We agree that, in principle, a fine-tuned 0.3B model could potentially regenerate the source document with noise (e.g., headers, footers, URLs) removed.
> > >
> > > In our preliminary experiments, we explored a simple document refining approach: training the model to directly generate high-quality fragments from raw documents, which is very similar to your proposal. Through this exploration, we observed the following, which leads to our current design:
> > >
> > > 1. **Limited capacity of small models**: Small models (e.g., 0.3B) struggled to perform fine-grained and document-specific tailoring, especially in the presence of diverse and noisy web content, while it is relatively easier for them to locate thses noises by line number or keywords.
> > > 2. **Efficiency concerns**: Generating clean fragments directly is often much less efficient in terms of both quality and cost. Specifically, this strategy leads to a larger number of output tokens, increasing computation costs. In contrast, identifying and removing noisy segments (e.g., headers, boilerplate, low-content sections) via code execution is more lightweight and cost-effective.
> > > 3. **Limited gain during experiments**: We also experimented with training ~1B models on both raw and fragment-refined data. Interestingly, the downstream performance of LM-refined data was similar to, or sometimes even slightly lower than, that of models trained on raw data, again demonstrating the first point.
> > >
> > > These observations motivated our decision to frame ProX as a modular refining framework that operates by identifying and filtering low-quality content, rather than regenerating full cleaned documents based on a small language model.

---

### Official Review · Reviewer_pU7q · 2025-03-21

**Overall Recommendation:** 4

**Summary:**

Data curation for LLMs typically relies on rule-based filtering to discard documents or refine them. However, these rules are inflexible and cannot adapt to the unique characteristics of each sample, but it would be laborious for a practitioner to determine how to refine/discard at the sample level. This paper proposes ProX, a procedure where a small LM is trained to produce a set of functions that discard and refine each sample. These functions are executed to refine the dataset, and then a LLM is trained on this curated dataset. This approach allows for per-sample flexibility by using the small LM, resulting in better performance than rule-based filtering approaches. Moreover, ProX's curation process is fairly efficient due to the size of the small LM and the programmatic use of the LM.

**Claims And Evidence:**

- Claims around ProX's performance and efficiency are supported by clear empirical evidence.
- The paper seems to claim that ProX's key advantage over prior heuristic-based approaches is its flexibility to refine each sample individually, in contrast to low-coverage, hard-coded rules. However, when comparing ProX with other filtering rules like FineWeb, Gopher, or C4, it's unclear whether ProX's improvements are due solely to its per-sample, LLM-based programming approach, or also because of the nature of the rules it uses. For example, ProX uses soft rules like: 1) discarding documents with low educational content or poor formatting scores 2) within chunks, removing elements such as navigation bars, URLs, and footers. In contrast, FineWeb/Gopher/C4 use different rules; for instance, C4 removes documents where the fraction of lines ending in punctuation is low. To better understand where ProX's performance gains come from, it would be insightful to run a baseline where you "convert" the traditional heuristic rules from FineWeb/Gopher/C4 into flexible, LM-driven programs that can adapt per sample. This would help isolate whether the improvement is due to better rules or the adaptive, per-sample application of those rules.

**Essential References Not Discussed:**

None.

**Experimental Designs Or Analyses:**

Experimental design and analyses appear sound.

**Methods And Evaluation Criteria:**

Proposed method and evaluation makes sense.

**Other Comments Or Suggestions:**

Typos:
- In table 1, "orignal" -> "original"

**Other Strengths And Weaknesses:**

Strengths:
- Remarkable results showing that the this approach can be used on a variety of model sizes and datasets, outperforming hard-coded rules as well as model-based data selection algorithms.
- ProX is an efficient curation approach (figure 7)---the total FLOPS used by ProX (including training) is less than standard training (especially for larger models). This is in contrast to other curation strategies, which are oftentimes equally or more expensive than the training itself.

Weaknesses:
- The applicability of ProX could be enhanced if the paper included more concrete information on how one (i.e., a domain expert) could specify new functions---I wonder if there are limitations of using Llama-70B to annotate the seed data for domain-specific functions. It would also be nice if approaches like Rephrasing the Web/WRAP (Maini et. al.) could be expressed in this framework and combined with the functions in this paper.

**Questions For Authors:**

1. Can the exact rules for C4, Gopher, and FineWeb be made available in the appendix?
2. Could experiments/analyses be conducted to isolate ProX's performance improvement due to per-sample application versus its particular rules?
3. Could the paper provide an example of how a practitioner/domain expert would add more complex functions to ProX, and which parts of the procedure they would need to adjust?

**Relation To Broader Scientific Literature:**

This paper strikes a middle ground between two existing types of approaches for data curation: 1) manual filtering rules that can cheaply be run across many samples, but are inflexible and have low coverage, and 2) expensive, per-sample LLM-based approaches, such as getting each sample to be scored/edited/synthesized by a powerful LLM. The paper shows that we can combine strengths from both of these paradigms to produce data curation methods that are relatively efficient yet of higher quality than heuristic approaches.

**Theoretical Claims:**

Not applicable.

---

> ### Author Rebuttal · Authors · 2025-03-31
>
> Thank you for recognizing our work! We are truly delighted to see your appreciation of ProX's novelty, efficiency, and effectiveness in improving data quality. Regarding your question:
>
> $\textrm{\color{blue}Question 1}$
> ```
> Can the exact rules for C4, Gopher, and FineWeb be made available in the appendix?
> ```
>
> Thank you for the suggestion. For these rule-based filtering methods, we primarily refer to the implementations provided in the FineWeb codebase. We are happy to summarize them below and will include the full details in the appendix in our revised version.
>
> - C4: We re-implement the C4 filtering rules, including:
>   1. Filtering lines with overly long words or too few words
>   2. Removing citations, placeholder text (e.g., lorem ipsum), JavaScript, bracket content, and web policy mentions
>   3. Discarding documents with too few sentences
> - Gopher: We adopt Gopher’s rules, including:
>   1. Minimum and maximum word count per document
>   2. Symbol-to-word ratio
>   3. Average word length per document
>   4. Heuristics such as bullet point filtering
> - FineWeb: We use FineWeb’s official implementation, which includes:
>   1. Punctuation ratio checks
>   2. Duplicate line removal
>   3. Filtering based on newline character frequency
>   4. Additional combined filters integrating the above rules
>
> We hope this clarifies our reference to these rule-based methods, and we’ll ensure their presentation in the appendix for reproduction.
>
> $\textrm{\color{blue}Question 2}$
> ```
> Could experiments/ analyses be conducted to isolate ProX's performance improvement due to per-sample application versus its particular rules.
> ```
>
> Regarding the performance of ProX in isolation, we would like to clarify two points:
> 1.  ProX inherently and functionally covers other rule-based filtering methods, as it leverages a language model to learn and apply data selection criteria, effectively capturing patterns encoded by handcrafted rules.
> 2.  During our preliminary study, we explored training the refining model using data filtered by FineWeb rules as negative samples. Compared to using the original raw data(a subset of Red-Pajama-V2), ProX achieved approximately **+1.0%** improvement in downstream performance, indicating its ability to identify lower-quality documents.
>
> Furthermore, we observed that more aggressive filtering tends to yield better performance. The use of quantifiable scores such as Edu Score and Format Score offers us the flexibility to adjust filtering thresholds and control the trade-off between data quality and quantity.
>
> $\textrm{\color{blue}Question 3}$
> ```
> Could the paper provide an example of how a practitioner/ domain expert would add more complex functions to ProX and which parts of the procedure they would need to adjust?
> ```
> Regarding the addition of specific rules or filtering strategies, we believe it is important to distinguish between document-level and chunk-level refining:
> - At the document level, if a domain expert aims to remove biased content (e.g., safety issues or toxic data), they can simply annotate relevant keywords and mark the corresponding documents with a drop_doc label. This allows for efficient and targeted removal based on expert knowledge.
> - At the chunk level, domain experts may need to:
>   1. Identify specific types of noise with clear patterns — such as frequently repeated dates in forum pages, pervasive advertisement slogans, or unnatural line breaks that disrupt the reading flow.
>   2. Abstract the noise removal strategy into a reusable function — for example, leveraging regular expression (regex) syntax for flexible and customizable pattern-based filtering.
>   3. Use model or human annotation to create seed data, which can then be used to train a refining model tailored to the noise patterns of interest.
>
> We hope this explanation clarifies our approach.

---

> > ### Comment · Reviewer_pU7q · 2025-04-04
> >
> > Thank you for your response. I still am not completely satisfied regarding Question 2 though.
> > > ProX inherently and functionally covers other rule-based filtering methods, as it leverages a language model to learn and apply data selection criteria, effectively capturing patterns encoded by handcrafted rules.
> >
> > Yes, but based on the appendix, the rules that underly ProX (e.g. education score, removing navigation bars, URLs, and footers) seem to still differ from C4/Gopher/FineWeb rules.
> >
> > So I am wondering if we could look at what happens if the underlying rules of ProX are the same as the other rules. Here's an example of how I would imagine doing this; take a Gopher rule:
> > Gopher rule: "Minimum and maximum word count per document" --> this traditionally requires some hardcoded number (i.e., if # words < 10 then discard)
> > To do a ProX version of this rule, you could create the seed data by prompting the LLama-70b model to flag documents if they appear too long or too short, but the key thing is that you don't enforce a hard rule, like 10 words.
> >
> > This is still the thing I am most curious about in the performance of ProX.

---

> > > ### Author Response · Authors · 2025-04-07
> > >
> > > We are glad to see our response address most of your questions.
> > > For your follow-up questions:
> > > ```
> > > So I am wondering if we could look at what happens if the underlying rules of ProX are the same as the other rules. Here's an example of how I would imagine doing this; take a Gopher rule: Gopher rule: "Minimum and maximum word count per document" --> this traditionally requires some hardcoded number (i.e., if # words < 10 then discard) To do a ProX version of this rule, you could create the seed data by prompting the LLama-70b model to flag documents if they appear too long or too short, but the key thing is that you don't enforce a hard rule, like 10 words.
> > > ```
> > >
> > > Thank you very much. We think that aligning the underlying rules and comparing hard-coded vs. model-based filtering is a valuable angle. Also, as we posted in our last response, in our preliminary study, we already trained ProX using data filtered by FineWeb rules as negative examples. This setup mimics applying the same rules but lets the model learn a softer boundary instead of enforcing hard thresholds. ProX achieved around **+1.0%** improvement in downstream performance over using the rule-filtered data directly.
> > >
> > > We really appreciate your suggestion, and fully agree that a broader study aligning with other rule sets is a promising direction for future work. As you suggested, to further investigate this, we additionally conducted an experiment using FineWeb-style prompts to generate seed data for ProX training.
> > >
> > > We chose FineWeb for consistency with our preliminary experiments and manageable compute cost. Similarly, we use Llama-3-70B-Instruct to construct these seed data, and our prompt focuses mainly on: 1. Punctuation ratio checks 2. Duplicate line detection 3. newline character frequency. Any flag document will be thus dropped.
> > >
> > > We present the results below for your reference:
> > >
> > > | Methods                                            | Avg Performance |
> > > |----------------------------------------------------|-----------------|
> > > | Raw data                                           | 42.1            |
> > > | FineWeb Rules                                      | 42.8            |
> > > | ProX (train on hard rule generated seed data)      | 43.1            |
> > > | ProX (train on llama generated seed data)          | 43.0            |
> > > | ProX-D (trained on Edu & Format focused seed data) | 43.5            |
> > >
> > > We believe these results suggest that:
> > > 1. the choice of underlying rules matters (none of them outperforms what is used in ProX-D and ProX-D+C)
> > > 2. meanwhile, even a small model trained with modest fine-tuning demonstrates slightly better document quality assessment than purely rule-based approaches. (which is also one of the main observations we have in ProX)
> > >
> > > We hope these updated results can answer your questions and clarify our main focus in ProX.

---

### Decision · Program_Chairs · 2025-05-01

**Decision:**

Accept (poster)

**Comment:**

The paper introduces Programming Every Example (ProX), a novel framework for refining pre-training data for large language models (LLMs). ProX leverages a small language model to generate and execute fine-grained operations, such as string normalization, for each individual example at scale. The experiments demonstrate that models trained on ProX-refined data consistently outperform other baselines across various benchmarks, showcasing its effectiveness across different model sizes and pre-training corpora.

**Strengths and Weaknesses Highlighted by Reviewers:**
**Strengths:**
  - **Remarkable results** showing ProX's effectiveness across different model sizes and datasets.
  - **Efficiency** in data curation, with lower computational costs compared to other strategies.
  - **Flexibility** in refining individual samples, leading to better performance than rule-based approaches.

**Weaknesses:**
  - **Applicability** could be enhanced with more concrete information on specifying new functions.
  - **Comparison** with traditional heuristic rules needs further exploration to isolate performance improvements.
 - **Prior work discussion** needs to be extended to include other relevant curation methods

**Rebuttal and Discussion with Reviewers:**
The authors provided detailed responses to reviewers' questions, clarifying the rules used for filtering and the performance gains of ProX. They conducted additional experiments using FineWeb-style prompts to generate seed data for ProX training, showing that the choice of underlying rules matters. The authors also addressed concerns about the efficiency and robustness of ProX, highlighting its advantages over full-document generation methods.

**Recommendation:**
All things considered, this paper would be a valuable contribution to ICML due to its innovative approach to data refinement and its demonstrated effectiveness across multiple benchmarks. The authors have shown that ProX can significantly improve the quality of pre-training data while maintaining efficiency. However, we expect the authors to improve the writing, particularly in clarifying the relationship to previous works and the practical implementation of new functions, as suggested by the reviewers, especially 4XW9.